# NON-INVASIVE NEURAL DECODING IN SOURCE RECONSTRUCTED BRAIN SPACE

## ABSTRACT

Non-invasive brainwave decoding is usually done using Magneto/Electroencephalography (MEG/EEG) sensor measurements as inputs. This makes combining datasets and building models with inductive biases difficult as most datasets use different scanners and the sensor arrays have a nonintuitive spatial structure. In contrast, fMRI scans are acquired directly in brain space, a voxel grid with a typical structured input representation. By using established techniques to reconstruct the sensors' sources' neural activity it is possible to decode from voxels for MEG data as well. We show that this enables spatial inductive biases, spatial data augmentations, better interpretability, zero-shot generalisation between datasets, and data harmonisation.

## 1 INTRODUCTION

Real-time non-invasive neural decoding is typically done using sensor measurements, with models translating their time series to useful predictions like which word is read (Duan et al., 2023) or picture is seen (Benchetrit et al., 2023). Raw non-invasive sensor data resembles a static random point cloud where each point has an associated time series, making the relation between sensor data points unintuitive. This renders tasks like developing data augmentations, interpreting decoding models, designing strong inductive biases, generalising between datasets, and combining them nontrivial. Many common deep learning modalities, such as images, do not suffer from this as they have a clear structure — they can be augmented by cropping, interpreted by masking regions, learnt using hierarchical local feature detectors, and combined by rescaling. Combining datasets is especially important as deep learning typically performs best with lots of data (LeCun et al., 2015).

These challenges are evident in existing works. Défossez et al. (2023) learn to predict perceived speech from magnetoencephalography (MEG) and electroencephalography (EEG) data, where despite their impressive engineering they do not combine datasets or use data augmentations. They do however use a spatial attention mechanism over the sensors' locations, thereby using spatial sensor space information. Jayalath et al. (2024) pre-train and fine-tune on different datasets, thereby requiring projecting their data into a learnt latent space which lacks explicit geometric structure. Moreover, it requires fine-tuning for datasets not seen during training. Afrasiyabi et al. (2024) learn a latent cross-modality representation but require corresponding functional magnetic resonance imaging (fMRI) and MEG/EEG data. Wang & Ji (2022) and Wang et al. (2023) transform sensors' time series to construct better input representations but treat each sensor independently. While some works use spatial data augmentations over sensors, these tend to be simple like swapping the left and right hemispheres or masking random channels (Rommel et al., 2022).

Other neural decoding modalities are amenable to more deep learning techniques as their inputs are spatially structured. fMRI scans are processed as 3D voxels, allowing the use of 3D convolutional networks (Mzoughi et al., 2020), data augmentations (Chlap et al., 2021; Ren et al., 2024), interpretability (Zintgraf et al., 2017), and transfer learning by combining datasets (Wen et al., 2023). Moreover, fMRI has a higher spatial resolution ($\sim$ 1mm) than MEG and EEG (several millimetres). However, its slow sampling speed ($\sim$ 0.5Hz) and bulky equipment requirements make it difficult to use for real-time decoding (Benchetrit et al., 2023). Invasive approaches have high spatiotemporal resolutions and are spatially structured but require subjects undergoing surgery, thereby limiting their use (Peksa & Mamchur, 2023). MEG and EEG can record brain signals at cognitively relevant timescales (on the order of ms) while being non-invasive. So far, these data are not typically decoded

in voxel space, possibly due to this mapping between measurements and their spatial sources being ill-posed and nontrivial (Mattout et al., 2006).

Still, approximate MEG and EEG source reconstruction is possible. This converts the sensor data to a 3D voxel grid of neural activity at different points in the brain, known as brain or source space. Being a regular grid with an underlying physical meaning source space has an inherent and intuitive spatial structure. This allows it to benefit from research on learning for 3D images (Maturana & Scherer, 2015) in general and medical imaging (Lu et al., 2019) in particular.

There is also a naturalness argument for why neural data should be analysed from a source instead of sensor perspective. Although representation learning should make different input representations equivalent, in practice structured inputs yield better performance as they enable data augmentations (Park et al., 2019; Shorten & Khoshgoftaar, 2019) and architectures with inductive biases (LeCun et al., 1995; Gilmer et al., 2017; Ho et al., 2020). In well studied modalities input representations that fit human perception tend to work best. Bitmap images are standard inputs in computer vision and not spatial gradient information, which were prevalent in pre-deep learning handcrafted features (Lowe, 2004; Dalal & Triggs, 2005). In speech processing log mel spectrograms — a logarithmic scale that approximates the human auditory response — are the de-facto standard input representations (Choi et al., 2018; Park et al., 2019). On the other hand, text has complex dependencies that make it difficult to design effective data augmentations, with state-of-the-art models often not using any (Vaswani et al., 2017). Structured inputs are also easier to interpret — practitioners care which brain regions play a role in decoding stimuli more than whether the most important sensor was number 15 or 32. While neural activations are not perceived in a sensory sense, they describe an underlying biological process instead of a measurement thereof.

Decoding from MEG/EEG source space has been done before, but previous works are either impractical for real-time decoding or do not use deep learning. For example, Daly (2023) rely on simultaneous fMRI recordings, Westner & King (2023) use only linear models and either simulated or small datasets, and Westner et al. (2018) use only random forests. Still, their results are promising — Westner et al. (2022) remarks that source reconstruction can induce denoising and for Westner & King (2023) decoding from sources outperforms sensors.

In this work we compare decoding in source to sensor space, with an emphasis on large-scale deep learning. After tuning a preprocessing pipeline (section 2) we show that source space' structure can help decoding, enabling models with inductive biases (3.1), spatial data augmentations (4), interpretability (5), zero-shot cross dataset generalisation (6), and learning from a combination of datasets without requiring projecting to a shared learnt latent space (7), which is sometimes known as "harmonisation" (Cheng et al., 2024). We are unaware of other works that accomplish the last two points.

## 2  Preprocessing and Source Reconstruction

Data is mapped from sensor to source space using source reconstruction (Hämäläinen et al., 1993), a set of techniques for determining neural activity from the sensors' electromagnetic measurements. These require anatomical brain scans to model how the fields permeate through their surroundings. For simplicity we focus here on MEG data although EEG source reconstruction is also possible.

As MEG data has many sources of noise a well-tuned preprocessing pipeline is important, especially for good approximate source reconstruction. While the pipeline can be designed by changing one of its settings, training a model on it, and then measuring decoding accuracy, this amounts to an expensive high-dimensional hyperparameter optimisation. Instead, we first train logistic regression classifiers on a large random sample of settings and then ablate the best ones,

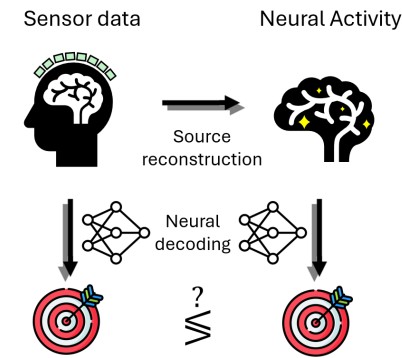

Figure 1: General setup — MEG data is measured using sensors, preprocessed, and used for decoding. Here we use source reconstruction to learn over estimated neural activity instead.

with both cases using a simple decoding task, here being heard speech detection — whether or not the subject is hearing spoken language. Other common tasks like phoneme feature classification (Jayalath et al., 2024) or decoding heard speech (Défossez et al., 2023) are not used as it is harder to get a meaningful signal for them. Different pipelines' outputs were also visually inspected to ensure their results are sensible, example shown in Appendix A. The settings, their meanings, and final values are given in Table 1.

Table 1: Preprocessing pipeline parameters and their final values.

| Stage | Parameter | Description | Final Value |
|---|---|---|---|
| **Sensor Preproc.** | Highpass frequency | Sets the signal's lower frequency limit using a bandpass filter | 0.1Hz |
| | Lowpass frequency | Sets the signal's upper frequency limit using a bandpass filter | 48Hz |
| | Downsampling frequency | Reduces the signal's sampling rate | 150Hz |
| | Notch filter | Whether to remove powerline frequencies | False |
| **Source Reconstruction** | Reconstruction method | Source reconstruction algorithm | Min. norm |
| | Voxel size | Voxel's dimensions | 15mm |
| | SNR | Assumed signal-to-noise ratio for source reconstruction regularisation | 3 |
| | Noise covariance form | Used for whitening data prior to source reconstruction | diagonal |
| | Voxel type | Whether to keep all activity components in voxels (vec) or only the field's magnitude (mag) | vec |

Source reconstruction usually uses a subject's anatomical scans, known here as structurals, to better model the relation between the sensor measurements and their sources. We use the Armeni et al. (2022) dataset for tuning the pipeline as it has such scans and multiple recording sessions per subject, allowing single-subject decoding. It contains MEG measurements of people listening to stories in English. For tuning the pipeline we train on the first subject's first session's recording while validating on the second session and testing on the third, each being 1-2 hours long.

The logistic regression classifiers showed several trends. For the sensor preprocessing a) lower highpass and lowpass frequencies tend to perform better, and b) the downsampling frequency and whether a notch filter is applied do not have a meaningful effect. Lowpass frequencies as low as 10Hz were found to perform well, surprising given that meaningful auditory processing occurs at higher frequencies (Hyafil et al., 2015). On the other hand much of the auditory response is low frequency, see Figure 4. Thus, as this is possibly due to logistic regression not having representation learning and hence latching onto coarse features we chose to use a lowpass frequency of 48Hz for the final pipeline, below the power line frequencies (50Hz for European and 60Hz for American measurements) but higher than typical neural auditory processing (Hyafil et al., 2015).

The source reconstruction parameters also exhibited some patterns. Tomographic (global) source reconstruction methods that assume correlated voxels (minimum norm (Hämäläinen & Ilmoniemi, 1994), sLORETA (Pascual-Marqui et al., 2002), and dSPM (Dale et al., 2000)) outperform a local one which does not (LCMV beamformer (Van Veen et al., 1997)). Vector source estimates — keeping the source activity's 3 vector components — perform better than magnitude based ones — taking only its magnitude. No typical covariance matrix, signal-to-noise ratio, or voxel size are significantly better than any others. Voxel sizes of 10, 15, and 20mm had similar results, likely due to MEG's poor spatial resolution as larger voxels may yield a higher signal-to-noise ratio as a larger volume is averaged over. Smaller voxels were not used due to memory constraints.

Given this, the best value was chosen for sensitive parameters while the rest were given typical default values, yielding a standard MEG preprocessing pipeline (Gross et al., 2013) with a few notable exceptions. The sensor data is lowpass filtered at 48Hz and downsampled to 150Hz to reduce the amount of data while preventing aliasing by having the lowpass filter be about 50% less than the Nyquist frequency. Although the covariance matrix' form seems insignificant, later larger-scale experiments showed that using a diagonal covariance yields slightly better results. Vector

Table 2: Ablations for parameters that reduce the input's information content. Frequencies are in Hz, balanced accuracies are reported. Parameters used for the final pipeline are marked in gray. In most cases methods that minimally decrease the input's information work best, with 2c finding little filtering preferable over none.

(a) Dimensionality reduction methods. PCA is per voxel so 3 components amounts to doing nothing.

| Method | acc |
|---|---|
| None | 66.9 |
| PCA (1 component) | 65.4 |
| PCA (2 components) | 65.0 |
| Parcels | 50.5 |

(b) Voxel type, whether to keep the reconstructed field as a vector or take its magnitude.

| Voxel type | acc |
|---|---|
| vec | 66.7 |
| mag | 62.3 |

(c) Highpass frequency, signals with a lower frequency are attenuated.

| Highpass freq | acc |
|---|---|
| None | 61.1 |
| 0.1 | 67.1 |
| 0.5 | 64.0 |
| 1 | 60.5 |

source estimates were chosen instead of magnitudes to make the sensor-to-source transformation be a simple linear transform per subject and to not discard information.

As preprocessing takes several minutes per subject the final processed data is cached. This makes sensor and source space dataloading times similar, with training time chiefly dictated by the batch size.

The final pipeline yields $400 - 900$ voxels, with the exact number depending on the subject's anatomy. Although decoding from a variable number of voxels is possible, fixed-domain decoding reduces information loss by not requiring domain-agnostic pooling. For multi-subject decoding source activity estimates are morphed into a standard template brain using both the original subject's and the template's anatomy, as per Avants et al. (2008). Usually this increases the number of voxels.

The template brain has $\sim 900$ voxels with three components each, giving an input dimension of $\sim 2700 \approx 10\times$ the number of sensors. Although MEG/EEG data is known to be low rank (Baryshnikov et al., 2004) and vector source estimates are assumed to have only two relevant components,[1] we did not use dimensionality reduction techniques such as applying PCA to a voxel's channels or grouping meaningful regions using parcellations (Eickhoff et al., 2018). This is to prevent inputting biases that may or may not help but to have the models learn which information is relevant, as often works best (Sutton, 2019). Indeed, Table 2a shows that dimensionality reduction harms performance.

## 3 SOURCE VS SENSOR SPACE

To test how well the representations compare we benchmarked both when decoding across a single and several subjects. The former uses the Armeni dataset as it has only three subjects but 10 hours for each, while the latter uses the Schoffelen et al. (2019) dataset due to it having many subjects but each having only one session ($\sim$1 hour). Schoffelen includes Dutch subjects listening to Dutch sentences with randomly shuffled words.

For Schoffelen subjects A2002, A2003 are used for validation, A2004, A2005 for testing, and A2006-A2010 for training. Armeni is split into 10 different recording sessions, with sessions 001-008 here being used for training, 009 for validation, and 010 for testing. This ensures the models have to temporally extrapolate as we found that randomly splitting the data can erroneously yield much higher performance due to leakage.[2] Only internal baselines are used due to the custom task,

---

[1]This is because the electric dipoles are usually assumed to be normal to the cortical surface (Bonaiuto et al., 2020).

[2]This is similar to Yang et al. (2024)'s data splits except they used training sessions after their validation/test sets and a dataset without structurals.

data splits and to minimise the setup's complexity.[3] All models are trained in a supervised manner on heard speech detection with a binary cross-entropy loss.

To test only the input representation's quality while minimising confounding factors, eg. inductive biases benefiting one domain more than the other, we train three-layer multilayer perceptrons (MLPs) with dropout. For the multi-subject case 16-dimensional subject embeddings are concatenated before each fully connected layer. This was found to work slightly better than additive embeddings or concatenating only before the first layer. For new "unknown" subjects the average embedding is used.

To minimise the optimisation's effects a random hyperparameter search was used for each setup, with results detailed in Appendix C. Unless stated otherwise all the following models have 0.25M parameters for multi-subject decoding and 0.5M for single-subject, with the latter having more training data.[4] We measure balanced accuracies as the data has different amounts of gaps/silence relative to speech. All models were given single time slices as inputs instead of short windows as we focus on comparing different input spatial representations. This is another reason heard speech detection is a convenient task — it requires far less temporal context than, for example, phoneme or word classification. Single time slices have the added benefit of speeding up training relative to using context windows.

As the models have similar capacities, equal inductive biases, and no additional information other than the input is used we a priori expect source and sensor space to perform similarly. Table 3 (middle) shows that for single subject decoding sensor space slightly outperforms source space, possibly due to the lower input dimensionality making the optimisation easier.

Table 3: Balanced accuracies for single-subject (Armeni) and inter-subject (Schoffelen) source and sensor space decoding. Errors denote standard deviations over subjects and $\geq 3$ random seeds throughout the paper.

| Input | Single-subject (Armeni) | | | | Inter-subject (Schoffelen) |
| | Subject 1 | Subject 2 | Subject 3 | Average | |
| --- | --- | --- | --- | --- | --- |
| Source | $66.33 \pm 0.09$ | $66.40 \pm 0.13$ | $67.54 \pm 0.18$ | $66.8 \pm 0.6$ | $53.5 \pm 0.5$ |
| Sensor | $67.44 \pm 0.07$ | $66.63 \pm 0.02$ | $68.21 \pm 0.09$ | $67.4 \pm 0.7$ | $54.0 \pm 0.4$ |

For inter-subject decoding the result is similar, as seen in Table 3 (right). Sensor space' inter-subject probability of improvement (Agarwal et al., 2021) over source space is 73%. Inter-subject accuracies are typically lower than single-subject as it is easier to generalise across time than across subjects (Csaky et al., 2023; Jayalath et al., 2024). Even for single-subject decoding accuracies are low relative to classical deep-learning modalities due to the task being hard and data being noisy.

Sensor and source performing similarly goes against some previous studies that found source space to work better, eg. Westner & King (2023); van Es & Schoffelen (2019). As they use either regular statistical methods or models with no representation learning, it is unclear if their performance gains are inherently from source space as a modality or due to its increased dimensionality.

The multi-subject setting is more common for large-scale non-invasive decoding due to having more data and allowing inter-subject knowledge transfer. Thus, unless specified otherwise, from here on all experiments are inter-subject.

## 3.1 USING SOURCE SPACE FOR INDUCTIVE BIAS

In practice the inputs' structures would be used to improve model performance. For source space a simple way to do so is using a 3D CNN, however this is difficult as the voxels do not form a cubic grid due to the brain being non-cubic. A naïve workaround is inscribing them into a larger 3D box

---

[3]This is not uncommon in neural decoding, with many important works making their own baselines (Hollenstein et al., 2019; Défossez et al., 2023; Wang et al., 2023; Jayalath et al., 2024; Afrasiyabi et al., 2024). While clearly suboptimal, it is a symptom of a maturing field without standardised MNIST-like setups.

[4]For reference, (Défossez et al., 2023)'s models have about 9M trainable parameters when trained on (Gwilliams et al., 2023)'s dataset, ∼6 times as much data as a single Armeni subject. In the current setup larger models gave similar results but were harder to train, likely due to their lack of inductive biases.

with the voxels outside of the brain being zero, which is what we do here. We train a 3D CNN with two squeeze-excite (SE) blocks (Hu et al., 2018) followed by two fully connected layers, with channelwise and regular dropout applied respectively before each.

We compare this to a model that can process irregular domains, a Graph Attention Network (GATs) (Veličković et al., 2017). GATs have seen success in other neuroscience tasks like creating cortical parcellations, albeit using MRI data (Cucurull et al., 2018). We use two GAT layers followed by two fully connected layers, with dropout applied similarly to the CNN. We use a regular nearest-neighbours graph, leading to most voxels having 6 edges.

These models have translation/permutation symmetries respectively, which does not reflect the brain's structure, eg. with some phenomena occurring only in one hemisphere. To break this symmetry positional embeddings are used, resulting in 6 input channels — 3 for the vector components and 3 for the $x, y, z$ positions. Subject embeddings are not added to the positional embeddings. Subjects not seen during training use the average subject embedding.

To see if these biases help sensor space as well a GAT was trained on it as well, with the graph depending on the sensors' locations. Each sensor is connected to its 5 nearest neighbours so it has a similar average degree to the voxel graph.

Table 4: Inter-subject (Schoffelen) balanced accuracies for different models in sensor and source space. Inductive biases improve performance. The most geometric/spatially-tuned model — the CNN — performs best.

|  | Sensor | | Source | | |
|---|---|---|---|---|---|
| Model | MLP | GAT | MLP | CNN | GAT |
| Accuracy | $54.0 \pm 0.4$ | $53.3 \pm 0.7$ | $53.5 \pm 0.5$ | $54.5 \pm 0.3$ | $52.7 \pm 0.3$ |

In spite of its inefficient input representation the CNN outperforms all sensor and source space models while the GATs are worse than the MLP baselines. The GATs struggle to learn, having lower validation accuracies than other models while also requiring more compute. The CNN's success indicates source space' spatial information's importance.

## 4 SPATIAL DATA AUGMENTATIONS

Source space's spatial information also enables spatial data augmentations. This is especially important in neural decoding where data is expensive and scarce. While many works developed temporal, learnt, or sensorwise augmentations for MEG and EEG (Rommel et al., 2022), there are few spatial augmentations and none that we are aware of in source space. We compare a generic modality-independent augmentation – mixup (Zhang, 2017) — with a 3D space-specific augmentation which is similar to masking parts of an image, which we call slice dropout. Slice dropout works by applying dropout to random planes of voxels. We report the dropout probability used for each axis, where in practice about $3\times$ as many planes are dropped as it is applied along the $x, y, z$ axes independently.

Table 5 shows that while slice dropout boosts the MLP it degrades the CNN's performance. This could be due to neighbouring voxels being highly correlated, thereby making masking planes ineffective for models that rely on local information. This led us to try a different augmentation where random cubes are masked out for some fraction of the inputs, which turns out to be effective for the CNN.

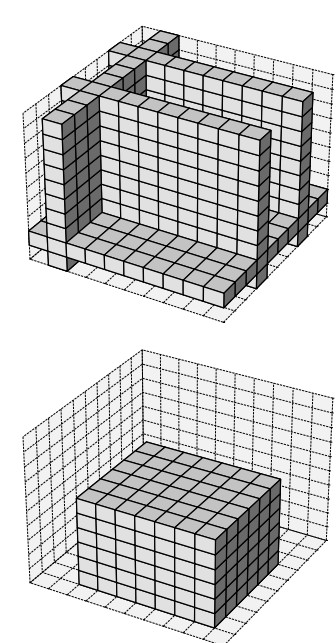

Figure 2: Illustration of slice dropout (top) and cube masking (bottom). Shaded voxels are set to zero.

Table 5: Balanced accuracies for different augmentations. Best results for each model are in bold. Parameters refer to the augmentation's defining parameter, where Cube Masking's $p$ is the percent of inputs it is applied to. $P_{\text{Best aug.}>\text{Baseline}}$ is the best augmentation's probability of improvement over its baseline.

|  | Sensor | Source | |
| --- | --- | --- | --- |
| Augmentation | MLP | MLP | CNN |
| Baseline | $54.0 \pm 0.4$ | $53.5 \pm 0.5$ | $54.5 \pm 0.3$ |
| Mixup $\alpha = 1$ | $53.7 \pm 0.4$ | $52.7 \pm 0.1$ | $53.7 \pm 0.2$ |
| Mixup $\alpha = 0.1$ | $\mathbf{54.3 \pm 0.4}$ | $53.3 \pm 0.6$ | $54.0 \pm 0.2$ |
| Slice dropout $p = 0.05$ |  | $\mathbf{53.9 \pm 0.4}$ | $54.5 \pm 0.4$ |
| Slice dropout $p = 0.1$ |  | $53.4 \pm 0.5$ | $54.7 \pm 0.4$ |
| Cube Masking $p = 0.5$ |  | $53.6 \pm 0.5$ | $\mathbf{54.9 \pm 0.4}$ |
| $P_{\text{Best aug.}>\text{Baseline}}$ | $76\%$ | $63\%$ | $76\%$ |

## 5 IMPACT OF DIFFERENT BRAIN REGIONS

To understand which parts of the brain the models rely on we evaluate them while masking different regions. We use the Harvard-Oxford atlas (Frazier et al., 2005; Makris et al., 2006; Desikan et al., 2006; Goldstein et al., 2007) to define regions of interest and test the single and inter-subject models that were trained without data augmentations. Due to a region's definition being sometimes ambiguous and the source reconstruction being imperfectly localised neighbouring voxels are included as a buffer, with only regions that have at least five voxels being considered. Regions are loosely divided based on their known functionality, with sources for each in Appendix D.

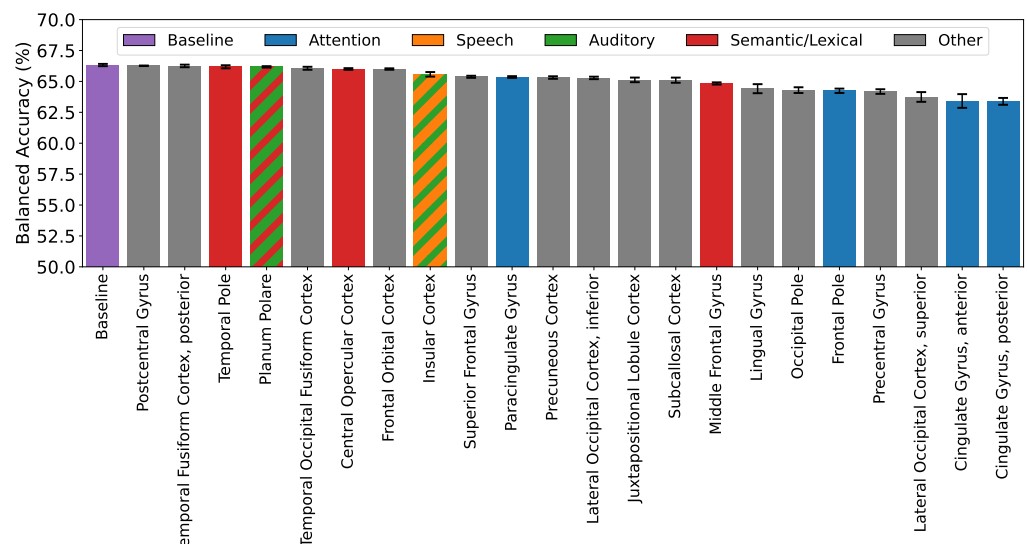

Figure 3: Accuracy when masking out different brain regions, shown here for the model trained on subject 001 in the Armeni dataset. Other subjects are shown in Appendix D. "Baseline" corresponds to no masking. For all single-subject models the baseline performs better than most masks. Errors are standard deviations over models trained on different seeds.

Surprisingly, dropping any specific region has little effect, with Figure 3 showing that the accuracy drops by at most ∼3% relative to the baseline. This could be due to the relevant activity being distributed, with the model not relying solely on any specific region, or because of technical reasons like the source reconstruction being insufficiently localised. A relevant region might not be included in the atlas as about a third of the voxels are unassigned. No simple consistent trend is seen across subjects. Better understanding this is left for future work.

## 6  ZERO-SHOT INTERDATASET GENERALISATION

Due to source space being a standard shared input representation it allows evaluating models trained on one dataset on subjects from another. Here we test how well the inter-subject (Schoffelen) models perform when evaluated on the single-subject (Armeni) data and vice-versa. Although all models are trained on heard speech detection the Schoffelen data is for native Dutch speakers listening to randomly shuffled Dutch words while Armeni subjects are native English speakers listening to English audiobooks. Moreover, each dataset has a different number and configuration of sensors, making cross-dataset sensor space evaluation impossible for fixed-domain models.

Each dataset has a slightly different setup when evaluated with the other's models. Armeni subjects are evaluated morphed into the template brain while Schoffelen test subjects are morphed into the relevant Armeni subject's anatomy. Test sets are the same as they are when evaluating in-domain.

Table 6 shows that the CNN performs comparably on both datasets, despite seeing only one during training, with other models being close to if not below random chance. Although the MLP and CNN have a small gap on the inter-subject test set the CNN generalises much better across datasets. Subject 003 is interesting as they exhibit bilateral neural activity unlike the other two subjects which are right-lateralised (Armeni et al., 2022). This may hint at why the CNN better generalises on this subject in contrast to the MLP — its convolutions have partial feature sharing,[5] thereby making the active hemisphere or activity's exact location matter less.

|  | Subject | | |
|---|---|---|---|
|  | 001 | 002 | 003 |
| **Sch.→Ar.** | | | |
| MLP | $51.5 \pm 1.6$ | $51.4 \pm 0.4$ | $51.0 \pm 2.0$ |
| CNN | $52.7 \pm 0.5$ | $53.5 \pm 0.7$ | $55.7 \pm 0.6$ |
| **Ar.→Sch.** (MLP) | $50.0 \pm 0.5$ | $51.2 \pm 0.2$ | $48.2 \pm 0.2$ |

Table 6: Inter-dataset evaluations. Sch.→Ar. indicates accuracies of models trained on Schoffelen (inter-subject) when evaluated on Armeni subjects. Ar.→Sch. accuracies are how well models trained on a specific Armeni subject generalise to the Schoffelen test set. Inter-subject models manage to generalise while single-subject ones fail.

Unsurprisingly, the single-subject models do not generalise across subjects. In spite of morphing the subjects' anatomies into the trained one's structurals the model likely overfits to its training data. It would be interesting to see whether morphing on the activity and not only the anatomical level allows the model to generalise, although this mapping is much more complicated.

## 7  LEARNING FROM COMBINED DATASETS

A common input representation also allows aggregating datasets. We add to the standard inter-subject training setup the first session from Armeni's subjects 001-002 while leaving the validation and test sets fixed. This gives an approximately equal number of hours per Schoffelen/Armeni subject in the training data. To see whether the additional data helps the inter-subject MLP and CNN's hyperparameters are optimised without changing their capacity. No data augmentations are used.

Table 7 shows that combining datasets helps given a model that can leverage them. The MLP struggles to learn with the extra data and performs worse, failing to transfer information between datasets. The CNN's performance improves, likely owing to better leveraging the additional data. Its probability of improvement with the combined data relative to the Schoffelen only baseline is 79%.

This is further supported by Table 8 showing that combining datasets boosts performance for all subjects using the same scanner. The better accuracy for subject 003 is somewhat surprising as subjects 001-002 exhibit right-lateralised neural activity while 003 is bilateral, thereby skewing the

---

[5]Only partial due to the positional embeddings breaking the translational symmetry.

Table 7: Learning from a combined dataset improves performance given a model with good inductive biases. Errors are standard deviations over seeds.

| Training data | **MLP** | **CNN** |
|---|---|---|
| Schoffelen only | $53.5 \pm 0.5$ | $54.5 \pm 0.3$ |
| Combined | $52.9 \pm 0.2$ | $54.8 \pm 0.4$ |

data away from bilateralism relative to the only Schoffelen baseline. Evidently the within-dataset similarities make up for inter-subject disparities.

Although subjects 001-002 have better performance for the combined data their variance increases significantly. It is unclear why this occurs.

Table 8: Armeni test accuracies for the CNNs trained on Schoffelen or the combined dataset. Subjects 001-002 are in the combined dataset while 003 is not. Combining data helps generalise across both seen and new subjects. Errors are standard deviations over seeds. $P_{\text{Comb.}>\text{Sch.}}$ is the empirical probability of improvement.

| | **Subject** | | |
|---|---|---|---|
| Training data | 001 | 002 | 003 |
| Schoffelen only | $52.7 \pm 0.5$ | $53.5 \pm 0.7$ | $55.7 \pm 0.6$ |
| Combined | $57.7 \pm 2.0$ | $55.9 \pm 2.1$ | $59.3 \pm 0.3$ |
| $P_{\text{Comb.}>\text{Sch.}}$ | 100% | 100% | 100% |

## 8 DISCUSSION AND FUTURE WORK

This work demonstrates source space' utility as a non-invasive neural decoding input representation instead of sensor data. Beyond simple performance improvements — either through enabling spatial inductive biases or domain-specific spatial data augmentations — it allows using techniques that decoding from sensors does not, namely model-agnostic cross-dataset evaluation and combination. Moreover, source space is more interpretable as it represents the object of interest, the brain, while sensor data is a proxy thereof.

Models that handle variable-size inputs, like transformers and graph neural networks, may carry these benefits over to sensor space. We are unaware of any works that try this and our attempt to train a graph attention network underperformed an MLP baseline. Computer vision still relies on fixed domain models although enough data allows relaxing that requirement (He et al., 2015; OpenAI, 2024), perhaps neural decoding will follow suit.

Still, source space has some clear limitations, eg. a higher input dimensionality. Although this was not a bottleneck here memory may become an issue when scaling to more subjects. Source reconstruction results in additional preprocessing but this is a one-off computational cost. Finding the right parameters for it is nontrivial, with our ablations being a step towards a standard pipeline. Unlike Westner & King (2023) we do not find source space' preprocessing to be prohibitively long as long as results are cached, with optimal batch sizes chiefly dictating a model's training time. As sensor space in some cases has better performance out-of-the-box it might be preferred in simple applications.

Source space could allow combining not only different MEG datasets but different neural modalities altogether, eg. EEG and fMRI. Given these domains' limited data cross-modal learning could enable valuable knowledge transfer.

## REPRODUCIBILITY STATEMENT

Anonymised code is attached as supplementary material, with a README explaining how to download the data and replicate different experiments. Preprocessing, hyperparameter optimisation, and model hyperparameters are given in section 2 and Appendix C. Appendix E has additional technical details on the training setup and benchmarking.

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

# APPENDIX

## A   PREPROCESSING VISUAL SANITY

Preprocessed data was visually inspected to ensure it has expected qualitative characteristics like higher activity after stimuli. This is visualised by epoching the data around the onset of speech, with an example in Figure 4. Both sensor and source data was visualised. The activity around $0.25s$ after the stimuli likely corresponds to speech processing — qualitatively akin to Capilla et al. (2013). Note this is different to general auditory processing, which is known to occur sooner (Charest et al., 2009).

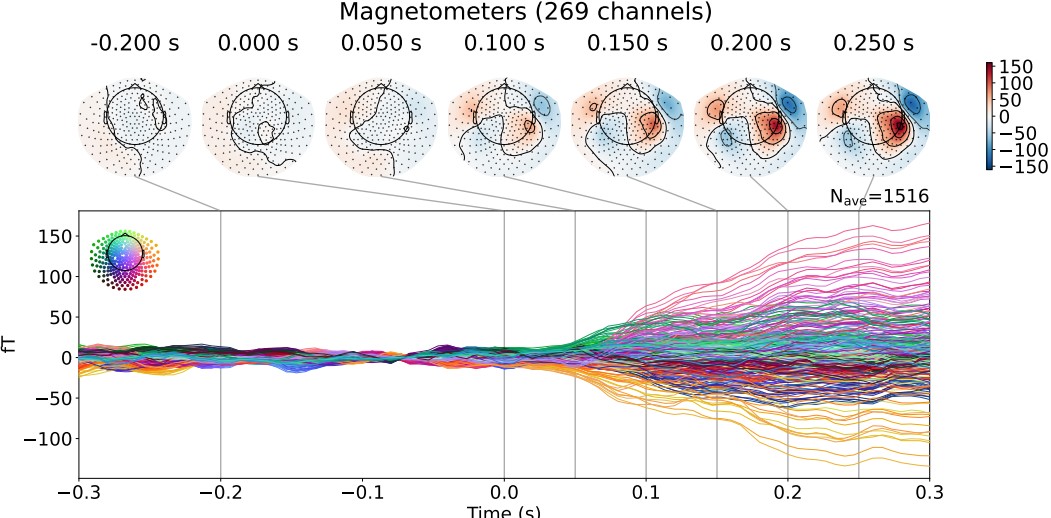

Figure 4: Epoched sensor activity for the final preprocessing pipeline. The topology activity maps are qualitatively similar to those found by Capilla et al. (2013). Averaged data for subject 1 session 1 in the Armeni dataset is shown.

## B   PREPROCESSING ABLATIONS

**Technical details.** The ablations' setup is similar to that used in the rest of the paper. Specifically, sensor data for the source reconstruction ablations uses the final preprocessing parameters given in Table 1. Both sensor and source data were standardised independently channelwise, with each session standardised independently and without any epoching, as detecting heard speech is a continuous task. The ablations' MLPs have two hidden layers with width tuned to have 0.25M parameters. The batch size is 128, learning rate is $10^{-5}$, and early stopping with a patience of 10 epochs and a minimum delta of 0.1% validation accuracy is used to terminate training. Test accuracies are given for the model with the best validation accuracy throughout training. Parcels were made using the Harvard-Oxford atlas, with each parcel having 12 features — the average, standard deviation, max, and min of its voxels' activities in each of the three axes.

**Results.** Ablations for all preprocessing parameters are given in Tables 9 and 10. Small variations in the final pipeline's accuracy, eg. 67.0% in Table 9a vs 67.1% in Table 9b, are due to random chance, with an estimated uncertainty of $\pm 0.3\%$. As these differences seem sufficiently small to not change the conclusions each ablation is run once.

In some cases values with a slightly lower accuracy were chosen for pragmatic reasons. For example, although 10mm voxels were slightly better than 15mm ones, the latter was preferred due to it resulting in $\sim$3x less voxels, enabling faster data loading and lower memory requirements. An SNR of 3 was chosen as it is the default in most implementations, eg. Sundaram & Lankinen (2024).

Table 10d's results are surprising as one would expect decoding to degrade given a different subject's anatomy. As for single-subject decoding this might not happen we ablated this further by

running the inter-subject MLP from section 3 when using a template structural directly instead of via the subject's anatomy. This had a slightly lower accuracy of $53.8 \pm 0.2\%$, showing that although structurals help they are not strictly required. The final pipeline used either a subject's or a template structural morphed into from a subject's structural, depending on if the experiment included a single or several subjects respectively.

Table 9: Ablations for sensor data preprocessing parameters. All frequencies are in Hz, balanced accuracies are reported. Parameters used for the final pipeline are marked in gray.

(a) Highpass frequency, signals with a lower frequency are attenuated.

| Highpass freq | acc |
|---|---|
| None | 61.1 |
| 0.1 | 67.1 |
| 0.5 | 64.0 |
| 1 | 60.5 |

(b) Lowpass frequency, signals with a higher frequency are attenuated.

| Lowpass freq | acc |
|---|---|
| 25 | 67.7 |
| 48 | 67.0 |
| 60 | 66.9 |
| 100 | 66.7 |
| 150 | 66.1 |

(c) Downsampling frequency has a minimal effect. Data is typically sampled at >1kHz.

| Downsampling freq | acc |
|---|---|
| 100 | 67.0 |
| 150 | 67.0 |
| 300 | 67.1 |

(d) Whether to attenuate signals at powergrid frequencies (50Hz for data used here).

| Notch filter | acc |
|---|---|
| With | 67.1 |
| Without | 67.0 |

Table 10: Ablations for source reconstruction preprocessing parameters. All frequencies are in Hz, balanced accuracies are reported. Parameters used for the final pipeline are marked in gray.

(a) SNR for local source reconstruction methods' regularisation.

| SNR | acc |
|---|---|
| 0.5 | 65.6 |
| 1 | 66.1 |
| 3 | 66.7 |
| 5 | 67.1 |

(b) Form of noise covariance matrix for source reconstruction.

| Noise cov. form | acc |
|---|---|
| regular | 65.2 |
| diagonal | 66.7 |
| scalar | 67.4 |

(c) Local source reconstruction methods outperform global ones.

| Source recon. method | acc |
|---|---|
| min. norm | 66.8 |
| dSPM | 67.0 |
| sLORETA | 66.8 |
| LCMV | 58.9 |

(d) Which anatomy to use for source reconstruction. Same template brain is used throughout.

| Structurals | acc |
|---|---|
| Template | 66.7 |
| Subj. | 66.7 |
| Subj.→Template | 66.4 |

(e) Voxel dimension in mm. All voxels are cubes with the specified side length.

| Voxel size | acc |
|---|---|
| 10 | 67.0 |
| 15 | 66.7 |
| 20 | 66.6 |

(f) Voxel type, whether to keep the reconstructed field as a vector or takes its magnitude.

| Voxel type | acc |
|---|---|
| vec | 66.7 |
| mag | 62.3 |

(g) Dimensionality reduction methods. Using none, thereby minimising information loss, works best.

| Method | acc |
|---|---|
| None | 66.9 |
| PCA (1 component) | 65.4 |
| PCA (2 components) | 65.0 |
| Parcels | 50.5 |

Similarly Table 9b shows that also given representation learning lower lowpass frequencies are better. This is likely due to the current setup where models process single time slices. Thus, to allow potential comparisons to settings with context and less temporally localised tasks we use the second-best ablated frequency, 48Hz.

## C  HYPERPARAMETERS

Random search distributions are shown in Table 11 with chosen hyperparameters given in Table 12 for single dataset models and Table 13 for combined dataset models. Unless stated otherwise, all models of a certain type for a given data split use these hyperparameters.

Table 11: Hyperparameters' random search distributions/values.

| Hyperparameter | Distribution/Value |
| --- | --- |
| Dropout prob. | $\{0.0, 0.1, 0.2, 0.3, 0.4, 0.5, 0.6\}$ |
| Learning rate | $10^{\mathcal{U}(-7,-3)}$ |
| Batch size | $\{16, 32, 64, 128, 256, 512, 1024\}$ |
| Max # epochs | 100 |
| Early stopping patience | 10 |
| Early stopping min. delta | $0.01\%$ |
| Weight decay | $10^{\mathcal{U}(-5,-0.5)}$ |

Table 12: Hyperparameters for inter-subject (Schoffelen) and single-subject (Armeni) models.

| Hyperparam. | Inter-subject | | | | | Single-subject | |
| --- | --- | --- | --- | --- | --- | --- | --- |
| | Sensor | | Source | | | Sensor | Source |
| | MLP | GAT | MLP | CNN | GAT | MLP | MLP |
| Dropout prob. | 0.1 | 0.2 | 0.1 | 0.6 | 0.2 | 0.5 | 0.2 |
| Learning rate | $5.4 \cdot 10^{-4}$ | $4.7 \cdot 10^{-7}$ | $5.4 \cdot 10^{-4}$ | $1.7 \cdot 10^{-6}$ | $8.5 \cdot 10^{-5}$ | $4.3 \cdot 10^{-5}$ | $7.0 \cdot 10^{-6}$ |
| Batch size | 16 | 128 | 16 | 256 | 256 | 256 | 64 |
| Weight decay | $1.7 \cdot 10^{-1}$ | $2.0 \cdot 10^{-5}$ | $1.7 \cdot 10^{-1}$ | $1.4 \cdot 10^{-5}$ | $7.6 \cdot 10^{-4}$ | $3.0 \cdot 10^{-5}$ | $9.8 \cdot 10^{-2}$ |

Table 13: Hyperparameters for combined dataset models.

| Hyperparam. | MLP | CNN |
| --- | --- | --- |
| Dropout prob. | 0.2 | 0.5 |
| Learning rate | $3.1 \cdot 10^{-6}$ | $2.1 \cdot 10^{-7}$ |
| Batch size | 512 | 16 |
| Weight decay | $2.8 \cdot 10^{-5}$ | $3.1 \cdot 10^{-3}$ |

## D  REGION MASKING

### D.1  REGION FUNCTIONS

Table 14 categorises each region's function. We stress that these are loose definitions as many regions' functionality is not fully understood.

### D.2  SUBJECTS 002 AND 003

Figures 5a and 5b show the trained model's accuracy when masking different regions for subjects 002 and 003 respectively.

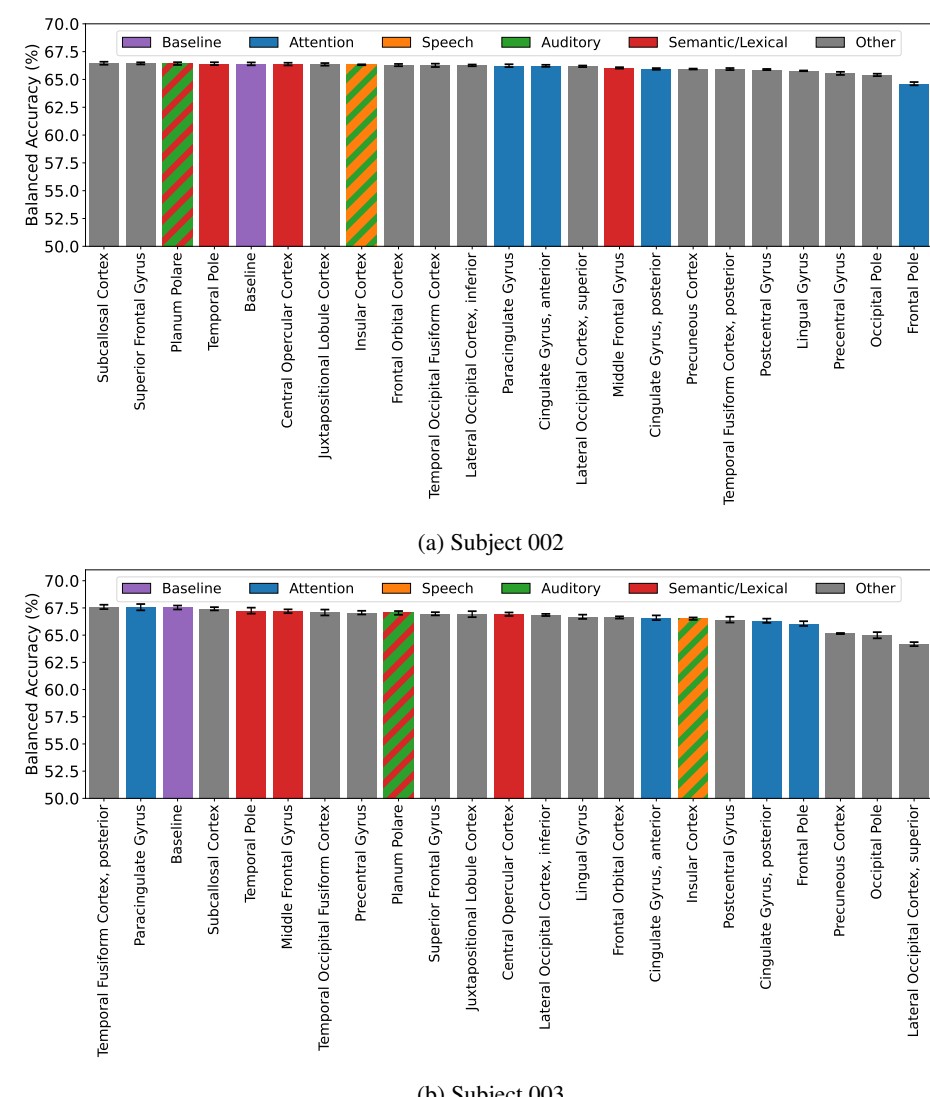

(a) Subject 002

(b) Subject 003

Figure 5: Accuracy when masking out different brain regions for (a) subject 002 and (b) subject 003.

## E   TECHNICAL DETAILS

**Setup**. All experiments were performed using PyTorch (Paszke et al., 2019). The preprocessing was done using MNE (Gramfort et al., 2014), MNE BIDS (Appelhoff et al., 2019), and FreeSurfer (Fischl, 2012). Experiments were run on a variety of GPUs, with the biggest using an Nvidia A100. No parallelism was used for training.

**Models**. Models were trained using AdamW (Loshchilov et al., 2017). Hidden dimensions were tuned to ensure models have the desired number of parameters. The SE blocks in the CNN have a reduction factor of 16 and use adaptive average pooling, with all other models having the same hidden dimension throughout their layers. One exception is the SECNN which has a pooling layer before two fully connected ones. The SECNN and GAT were chosen over similar variants, eg. a regular 3D CNN for the former and a graph convolutional network (Kipf & Welling, 2016) or simple graph convolution (Wu et al., 2019) for the latter, as they gave the best results. Positional embeddings were normalised to be in $[-1, 1]$ along each axis. Hidden dimensions are tuned so models approximately have the desired number of parameters.

**Benchmarking**. Many papers use the standard error of the mean instead of standard deviation, thereby showing errors which are not indicative of the setting's variance and are smaller when more seeds are used. We believe standard deviation is more appropriate here as it indicates a setup's consistency. A $\pm$ indicates at least 3 random seeds. More were used when employing statistical tests or by accident.

We use probability of improvement (Agarwal et al., 2021) instead of statistical comparisons like Welch $t$-tests as the former does not assume a normal distribution and is known to be robust under few samples. We found that Welch $t$-tests sometimes gave misleading results due to non-normal samples having a high variance. For example, there are 7 "Schoffelen only" subject 002 samples in Table 8 and 4 "Combined", with all "Schoffelen only" samples being less than the "Combined" ones. Due to the latter having a high variance a Welch $t$-test gives a $p$-value of $0.11$.

**Data augmentations**. Cube masking was done by randomly choosing two voxels in a grid and masking all points between them.

## F    BROADER IMPACTS

This work proposes an alternative input representation for neural decoding. Potential issues caused by advancing this technology are present here as they are in other works in the field, namely loss of privacy by enabling intrusive technologies, societal inequalities due to some having access to it while others do not, and so on. The potential benefits exist here as well, such as allowing those with speech impairments, eg. paralysed individuals, to communicate again.

## G    LICENCES

Armeni et al. (2022)'s dataset is distributed under a CC-BY-4.0 licence. Some preprocessing code is based on Défossez et al. (2023) which is distributed under a CC-BY-NC 4.0 licence. The Schoffelen et al. (2019) dataset is distributed under a RU-DI-HD-1.0 licence from the Donders Institute for Brain, Cognition, and Behaviour.

Table 14: Regions in the Harvard-Oxford atlas used for masking experiment. Only functions relevant to the task (hearing stories/speech) were categorised, with the categories being Attention, Auditory, Semantic/Lexical, and Speech

| Region | Function | Source | Note |
|---|---|---|---|
| Central Opercular Cortex | Semantic/Lexical | (Mălîia et al., 2018) | |
| Cingulate Gyrus, anterior | Attention | (Pardo et al., 1990) | |
| Cingulate Gyrus, posterior | Attention | (Weissman et al., 2004) | |
| Frontal Orbital Cortex | | (Rolls et al., 2020) | |
| Frontal Pole | Attention | (Kimberg & Farah, 1993) | |
| Insular Cortex | Speech, Auditory | (Uddin et al., 2017) | |
| Juxtapositional Lobule Cortex | | (Coull et al., 2016) | Known sometimes as the Supplementary Motor Area. |
| Lateral Occipital Cortex, inferior | | (Grill-Spector et al., 2001) | |
| Lateral Occipital Cortex, superior | | (Grill-Spector et al., 2001) | |
| Lingual Gyrus | | (Zhang et al., 2019) | Not associated with language processing. |
| Middle Frontal Gyrus | Semantic/Lexical | (El-Baba & Schury, 2023) | |
| Occipital Pole | | (Rehman & Al Khalili, 2023) | |
| Paracingulate Gyrus | Attention | (Gennari et al., 2018; Wysiadecki et al., 2021) | |
| Planum Polare | Auditory, Semantic/Lexical | (Deouell et al., 2007; Keenan et al., 2001) | |
| Postcentral Gyrus | | (DiGuiseppi & Tadi, 2023) | |
| Precentral Gyrus | | (Banker & Tadi, 2023) | |
| Precuneous Cortex | | (Al-Ramadhani et al., 2021) | |
| Subcallosal Cortex | | (Dunlop et al., 2017; Sobstyl et al., 2022) | |
| Superior Frontal Gyrus | | (Li et al., 2022) | |
| Temporal Fusiform Cortex, posterior | | (Cohen et al., 2000; Weiner & Zilles, 2016) | Associated with visual word recognition. |
| Temporal Occipital Fusiform Cortex | | (Weiner & Zilles, 2016) | |
| Temporal Pole | Semantic/Lexical | (Herlin et al., 2021) | |

