# OpenReview forum: "Non-invasive Neural Decoding in Source Reconstructed Brain Space"
_ICLR.cc/2025/Conference — Submitted to ICLR 2025_

### Official Review · Reviewer_PhyF · 2024-10-24

**Soundness:** 1
**Presentation:** 3
**Contribution:** 1
**Rating:** 1
**Confidence:** 5

**Summary:**

The paper investigates the use of source imaging for speech detection tasks. The authors compare the classification accuracy when using sensor space and source space as inputs across two datasets, employing three different network architectures. Additionally, the paper explores spatial data augmentation, cross-dataset training and testing for the two datasets.

**Strengths:**

* This is an important and beneficial topic for the field.
* The paper performed many experiments including cross validation, ablation studies, etc.

**Weaknesses:**

I have serious doubt regarding the validity of the result. This paper did not justify the use of source space signals over sensor space signals. Specifically: (1) There is no significant improvement in classification accuracy when using source space signals. (2) The additional advantages claimed for source space signals over sensor space are unsubstantiated, as sensor space signals can also support cross-subject learning and testing (although its accuracy is unknown).
While the benefits of source imaging over sensor space signals have been demonstrated in several fields, the authors did not apply source imaging correctly to biological signals, leading to little improvement from using source imaging.
1. Minimum Norm Estimation (MNE) is essentially ridge regression. Although it is a widely used algorithm for source imaging, its localization error is higher compared to more modern imaging methods, and it cannot provide an accurate spatial distribution for the source. The authors used the entire brain space as input, which makes an accurate estimate of the whole source space crucial, but MNE tends to produce numerous false positives.
2. Source imaging can improve the signal-to-noise ratio (SNR) for specific regions or signals of interest. However, the authors did not properly select the signal segment and used the entire source region. In comparison to sensor space, where the low-SNR input consists of a few dozen channels, the source space contains hundreds of voxels, making it more difficult to extract relevant information.
3. Figure 4 does not display a proper auditory evoked potential for expected MEG signal waveforms and source imaging results, as showed in dataset's source paper (Schoffelen, 2019)  Given that MNE is ridge regression, the regularization parameter (lambda), which is related to the SNR of the sensor space, plays a crucial role. However, since the authors used single time points as the dependent input, the SNR at each time point varies significantly.
4. The claim that the cross-dataset "zero-shot" approach is impossible in sensor space lacks proof. It is feasible to perform wrapping in source space to map each subject to a common space, and the same can be done in sensor space. The accuracy of both methods remains subject to evaluation.

**Questions:**

What is the input time segments?

---

> ### Author Response · Authors · 2024-11-15
>
> Thanks for the feedback, we would be happy if you could help us better understand some of your comments.
>
> **C1:** This paper did not justify the use of source space signals over sensor space signals. Specifically: (1) There is no significant improvement in classification accuracy when using source space signals.
>
> **Q1:** We are keen to understand \- why do the justifications given in the paper, both conceptual and empirical, feel insufficient?
>
> Conceptually, the first two paragraphs of the introduction explain why source space allows using techniques which are difficult if not, as far as we are aware, impossible in sensor space. Lines 62-75 give a high level motivation.
>
> Empirically, tables 4-5, show improvements over sensor space baselines, while tables 6-8 show generalisation across datasets/subjects that can’t be done in sensor space without projecting the data into a shared learnt (hence generally unstructured) latent space. While the accuracies are not high, this is typical for neural decoding and broadly difficult deep learning tasks, as mentioned in lines 248-251. For example, translation for low resource languages typically has a much lower BLEU score than for high resource languages (eg. German, English) \[1\]. Moreover, we use the probability of improvement as a statistical test to ensure that the results are significant and not a fluke, coupled with extensive hyperparameter tuning (mentioned in lines 224-228 and Appendix C) to decouple confounding factors.
>
> ---
> **C2:** (2) The additional advantages claimed for source space signals over sensor space are unsubstantiated, as sensor space signals can also support cross-subject learning and testing (although its accuracy is unknown).
>
> **Q2:** Cross-subject learning over sensor space is indeed possible, eg. we cite an example in lines 37-40. However this requires projecting the data to a learnt, generally unstructured, latent space, as mentioned in lines 85-86. Was this clear or should it be better emphasised throughout?
>
> Why do the advantages feel unsubstantiated? Eg. are the baselines unconvincing, and if so then why? Is it due to the accuracies, while being statistically significantly different, not being high relative to more classical deep learning?
>
> We agree the contrast with sensor space could be made clearer. We’ll add an explicit sensor space baseline to section 7, learning from combined datasets, where data is projected into a learnt shared space. Preliminarily we found such a model to achieve a balanced accuracy of 53.9+-0.4%, similar if not a bit worse than the one trained on only the Schoffelen sensor data, likely due to the MLP not managing to leverage the additional data owing to its lack of inductive biases, as was the case for the source space MLP. To ensure the comparison is not affected by a different amount of engineering put into source vs sensor space we’ll investigate more variants before adding the best one to the paper.
>
> We are unaware of established methods to project sensor space data into a shared space as there are for source space, but we discuss that properly in a later comment.
>
> ---
> **C3:** While the benefits of source imaging over sensor space signals have been demonstrated in several fields, the authors did not apply source imaging correctly to biological signals, leading to little improvement from using source imaging.
>
> **Q3:** We are extremely interested in this point as we took careful care to ensure we do source imaging properly. What in the given results indicates they were not used properly? The followup comments are addressed here:
>
> ---
> **C4:** Minimum Norm Estimation (MNE) is essentially ridge regression. Although it is a widely used algorithm for source imaging, its localization error is higher compared to more modern imaging methods, and it cannot provide an accurate spatial distribution for the source. The authors used the entire brain space as input, which makes an accurate estimate of the whole source space crucial, but MNE tends to produce numerous false positives.
>
> **Q4:** MNE was ablated against other source imaging methods, detailed in lines 148-152 and more extensively in Appendix B, and found to perform similarly. Although more sophisticated techniques exist MNE is still widely used, and there are well-cited studies arguing for it over other modern approaches, eg. \[2\]. Given these empirical and conceptual reasons, why would MNE be a poor choice here?

---

> > ### Author Response · Authors · 2024-11-15
> >
> > **C5:** Source imaging can improve the signal-to-noise ratio (SNR) for specific regions or signals of interest. However, the authors did not properly select the signal segment and used the entire source region. In comparison to sensor space, where the low-SNR input consists of a few dozen channels, the source space contains hundreds of voxels, making it more difficult to extract relevant information.
> >
> > **Q5:** This is an important conceptual point we believed we addressed in the paper, is it not clearly presented? Eg. parcellations \- using averaged brain segments instead of the voxel input, as is common in some source imaging studies, was ablated and discussed in Tables 2, lines 189-196, and Appendix B. It was empirically found to work worse than using the entire brain as input. Lines 189-196 give a conceptual reason why this may happening when using representation learning, as it goes against “The Bitter Lesson” in spirit \- we would end up throwing away potentially relevant information.
> >
> > As for the SNR, some preliminary works actually point to the opposite direction, with source space having a potentially better SNR, as mentioned in lines 252-255 (also discussing the dimensionality issue) and 79-81. As mentioned there, \[3\] found source space to have a better SNR than sensor space for their task and \[4\] found it to induce denoising.
> >
> > ---
> > **C6:** Figure 4 does not display a proper auditory evoked potential for expected MEG signal waveforms and source imaging results, as showed in dataset's source paper (Schoffelen, 2019\) Given that MNE is ridge regression, the regularization parameter (lambda), which is related to the SNR of the sensor space, plays a crucial role. However, since the authors used single time points as the dependent input, the SNR at each time point varies significantly.
> >
> > **Q6:** Are you referring to Figure 3 in \[5\]? Note that Figure 4 in this work is not for the source space but for the sensor space data, hence the topological activity maps with regions outside the subject’s head. As mentioned in lines 817-818, these results are akin to those seen in existing works. Should it be further emphasised that this is for sensor space? Are there reasons to believe the preprocessing is erroneous?
> >
> > We are unsure we understand, what’s the problem with the lambda parameter/SNR? As for the lambda parameter, it was briefly mentioned in lines 152-154 and thoroughly ablated in Appendix B. We found that an SNR of 3, corresponding to lambda=1/3^2 (which is the default in many libraries, eg. MNE Python), works well. Different points having a different SNR is of course unideal but would exist to some extent also in sensor space. Would this somehow invalidate any of the given results?
> >
> > ---
> > **C7:** The claim that the cross-dataset "zero-shot" approach is impossible in sensor space lacks proof. It is feasible to perform wrapping in source space to map each subject to a common space, and the same can be done in sensor space. The accuracy of both methods remains subject to evaluation.
> >
> > **Q7:** Could you please point towards any works or methods that do sensor space warping? This would be extremely interesting, we are unaware of any such works. Evidently, all dataset combination/harmonisation works we are aware of require a shared learnt space, hence being impossible for new datasets with different sensor configurations. As for evaluating zero-shot generalisation via source-space warping to a shared template brain, this is what section 6 is about. Was this unclear?
> >
> > What do you mean by “"zero-shot" approach is impossible in sensor space lacks proof”? Do you mean evidence? There are specialised models that would potentially be able to do so, which is discussed several times eg. in lines 273-278, 385-387, 464-469. Their disadvantages are given conceptually in lines 183-188 and empirically shown in Table 4, where the sensor GAT, a model that could in principle handle different sensor configurations as inputs, underperforms.
> >
> > In hindsight this point is not as clear and nuanced as we would have wanted it to be, so we will clarify it.

---

> > > ### Author Response · Authors · 2024-11-15
> > >
> > > **C8:** What is the input time segments?
> > >
> > > **Q8:** The model gets as input all the voxels at a certain point in time, is this what you were referring to? As mentioned in lines 229-232, this is why heard speech detection was chosen \- it allows focusing mostly on the spatial instead of temporal aspects of the representation.
> > >
> > > **References**
> > > \[1\] Arivazhagan, Naveen, et al. "Massively multilingual neural machine translation in the wild: Findings and challenges." *arXiv preprint arXiv:1907.05019* (2019).
> > > \[2\] Hauk, Olaf. "Keep it simple: a case for using classical minimum norm estimation in the analysis of EEG and MEG data." *Neuroimage* 21.4 (2004): 1612-1621.
> > > \[3\] Mats W.J. van Es and Jan-Mathijs Schoffelen. Stimulus-induced gamma power predicts the amplitude of the subsequent visual evoked response. NeuroImage, 186:703–712, 2019\. ISSN 1053-8119. doi: https://doi.org/10.1016/j.neuroimage.2018.11.029.
> > > \[4\] Britta U Westner, Sarang S Dalal, Alexandre Gramfort, Vladimir Litvak, John C Mosher, Robert Oostenveld, and Jan-Mathijs Schoffelen. A unified view on beamformers for m/eeg source reconstruction. NeuroImage, 246:118789, 2022\.
> > > \[5\] Jan-Mathijs Schoffelen, Robert Oostenveld, Nietzsche HL Lam, Julia Udd´en, Annika Hult´en, and Peter Hagoort. A 204-subject multimodal neuroimaging dataset to study language processing. Scientific data, 6(1):17, 2019\.

---

> > > > ### Comment · Reviewer_PhyF · 2024-11-17
> > > >
> > > > **C1 & C2**:
> > > > I acknowledge the conceptual significance, which I have highlighted as the strength of your paper. In Table 4, the reported improvement of Sensor MLP (54.0) over Source CNN (54.5)—a 0.5% increase—is either a direct head-to-head comparison (e.g., sensor 2D CNN vs source CNN) or statistically tested (I doubt it is statistically significant). Similarly, the comparisons in Table 5—between sensor space vanilla and source space model with spatial information and data augmentation—is not a fair comparsion.
> > > >
> > > > **C3–C5**:
> > > > As noted in my original comment, MNE (and all other tested methods) does not provide accurate spatial information; it provides location information rather than spatial extent. Yet, you are using methods like 3D CNN, which need an accurate spatial extent estimation of the source space to maximize its advantage. Additionally, SNR improvements depend on the type of noise being addressed. In the case of your use of source reconstruction results, it does not appear to increase the SNR.
> > > >
> > > > **C6 & C8**:
> > > > The visualizations are cut off at the peak, and the full evoked response is not presented. Could you clarify the time segment used in the training? Is it from -300 ms to 300 ms, or something else? Did you include any pre-stimulus data?
> > > >
> > > > **C7**:
> > > > I was referring to use interpolation to match channel differences between different montages. However, I realized that all the montages used in your study are CTF-275, which means inter-dataset performance could have been evaluated in the sensor space without requiring interpolation or wrapping to compare with the source space method.

---

> ### Author Response · Authors · 2024-11-17
>
> Thanks for your response, this clears some points up. Regarding each one:
>
> **C1 & C2:** In Table 4 the probability of improvement between the two cited models is 84% and the p-value when performing a Mann-Whitney U-test or Welch t-test (inequal variances) are 0.015 and 0.005 respectively - all three pointing to a statistically significant improvement. Note the standard deviations given in the table, not just the values - we believed the improvement was clearly sufficiently significant without citing the statistical tests but indeed for completeness we’ll add them. Do you still feel like this improvement is unfair or not scientifically sound? If so, why?
>
> As for Table 5, the comparisons are not between vanilla sensor space and source space with data augmentation but between data augmented versions of both. The sensor GAT model in Table 4 has spatial information, as noted in lines 284-286, but performs worse, hence why we used the MLP for the data augmentation comparisons. We agree a CNN-esque sensor space baseline is sensible and will update here once we have one.
>
> ---
> **C3–C5:** Ah, we believe we understand now. Just to make sure - you mean that tomographic/global methods, such as MNE, smear the resulting activity so it is akin to a blurred image instead of one with sharp details, where CNN-like models that roughly rely on hierarchical edge detectors are better suited for the latter instead of the former? If so, while we agree that intuitively local methods (eg. an LCMV beamformer) would be better suited then, as per the ablations detailed in section 2 and Appendix B it performs worse. Although better tuned source reconstruction techniques might work even better, this does not nullify the CNN’s empirical success given the current setup. This is intuitive - even if the input is analogous to a blurred image, a CNN would perform better on it than an MLP.
>
> As for the SNR, how did you reach the conclusion that our setup does not improve SNR? Please see comments and questions in the next reply. Moreover and more importantly, how would this invalidate any of the given results?
>
> ---
> **C6 & C8:** We would be happy if you could answer some of the questions in the previous comments, specifically:
> - Are there reasons to believe the preprocessing is erroneous?
> - We are unsure we understand, what’s the problem with the lambda parameter/SNR?
> - Regarding the discussion on the lambda parameter/SNR - would this somehow invalidate any of the given results?
>
> As for the new questions - as mentioned in the initial reply to C8 and in the places we cite there in the paper, the input is single time slices - not epoched/windowed data. Hence, there is no pre-stimulus data.
>
> As for Figure 4 - which period are you interested in? What is the underlying problem you are wary about? We are happy to generate a new plot but are especially interested in ensuring everything is scientifically sound.
>
> ---
> **C7:** We would be happy if you could send a paper that does anything like this as all works we are aware of and cited combine sensor space datasets using learnt methods. Do you mean spatially interpolating sensor values in one dataset based on the locations in another?
>
> As for all montages being CTF-275 - while technically true, the Armeni dataset has only 269 functioning sensors while the Schoffelen one has 273 for the subjects we used, and it is unclear if in both cases the coils were in the same location, making combining the two datasets in sensor space still nontrivial. Regardless and arguably more importantly, we do not see how this would change the conceptual point - the two datasets having similar setups doesn’t mean the problem of combining generic datasets doesn’t exist in general, or that this demonstration is invalid.

---

> > ### Comment · Reviewer_PhyF · 2024-11-17
> >
> > **C1 & C2**: Table 4 part: Thanks for the additional data, please add the statistical test result and please do not forget to add the sample size N.
> >
> > **C3–C5**: That is the intuition. But LCMV is not a good method for extent estimation either. For SNR, my intuition comes from the smeared source estimation. It does not invalid the result.
> >
> > **C6&C8**: I am not saying your preprocessing is wrong. Single time slices can come from time samples in -300ms - 300ms. It can also come from time samples in 250ms - 350ms. The source location could also be different between 100-200ms and 250-300ms. (See Schoffelen paper Fig 3.) Don't you think that is important to clarify for a scientific paper so that people can know what is your input? It would be good to plot the full epoch, where both positive and negative peak of an evoked potential are visible. Please also plot the source reconstruction.
> >
> > "single time slices" does to equal to having no pre-stimulus data. So no, you did not mention this before.
> >
> > You are dealing with a spatiotemporal signal. Why do you think more than one reviewer mentioned "temporal"? "What would discussing/experimenting with temporality add?" These time slices are not independent; they are often processed step by step for simplicity. In a paper utilizing MEG and source imaging, don’t you think it’s crucial to properly visualize the spatiotemporal signal you’re working with? Do you truly believe there’s no value in adding a discussion about the temporal properties of MEG?"
> >
> > **C7**: If only a few channels are different, mne-python and other toolboxes have channel interpolation functions. You are using MLP and CNN anyways. You are not taking into account (x,y,z) location of the input, so you do not need to consider the (x,y,z) location dis-match. I do not see why cross-dataset sensor test would be a problem.

---

> ### Author Response · Authors · 2024-11-18
>
> **C3–C5:** Thanks for the clarification. We are left a bit confused by this discussion, specifically how it relates to our work. In your original review you mention issues with the use of source reconstruction generally and MNE specifically, which we found surprising given both our knowledge of the field and the ablations we did. Following this discussion we understand the issues regarding MNE but still not why points 1+2 in the original review are weaknesses in this work. As they touch on the core scientific methodology, if following these clarifications you think there are no problems with the preprocessing/how source reconstruction is used here we would be happy if you could clarify that. Otherwise we want to ensure our method is sound. This general discussion is related to the next comment.
>
> ---
> **C6&C8:** For the first and second paragraphs - first thanks for the clarification on the preprocessing, we were worried you noticed an issue that snuck by us. We understand your confusion now about the input, we did not consider other options so thanks for the clarification, apologies for earlier misunderstandings. We will make this clearer in the main paper. The input is the single slice at the time of stimuli, so given that the sampling rate is 48Hz you can think of it as representing a [-0.01s,0.01s] window around it. This is a conscious design choice we’ll add to the appendix - Appendix A.4 in [1] showed that different delays/offsets do not have a big effect on the accuracy. For completeness we performed this ablation and indeed saw that the offset has a small if not negative effect, as seen in the following table. We will add it to Appendix B and mention it in the main text.
>
> | Label Offset (ms) | Test Accuracy (%) |
> |--------------------|-------------------|
> | -500              | 62.9             |
> | -400              | 63.3             |
> | -300              | 64.1             |
> | -200              | 65.1             |
> | -100              | 66.1             |
> | 0                 | 67.1             |
> | 100               | 67.1             |
> | 200               | 66.1             |
> | 300               | 64.9             |
> | 400               | 64.7             |
> | 500               | 64.7             |
>
> As for the sensor/source reconstruction evoked plots - note that Fig. 3 in the Schoffelen paper is the activity over time for parcellations, not the sensor data. We agree a more complete visualisation should be added, here are the plots (https://filebin.net/k3fy6v0ce6in6ydg). The sensor measurements over time in a [-0.6s,0.6s] range, where the negative peak is also visible. The voxel data over time shows each vector component in each voxel being plotted separately. Thanks for the suggestions, they will both be added to the paper.
>
> As for the last paragraph - we believe reviewer bKqP is discussing a different kind of temporality, specifically on tasks that naturally require longer contexts eg. word classification. As mentioned in the previous reply and as we did here, we are happy to create the extra plots as they help make the paper more scientifically sound. Here we compare different spatial representations and hence focus on them, as mentioned in lines 228-230, with temporality being important but orthogonal to our work. Do you feel that this discussion is insufficient? If so, why?
>
> ---
>
> **C7:** We first touch on the conceptual issue which we are confused by and then on the technical point. Even assuming one can easily do the zero-shot cross-dataset sensor model evaluations in this specific instance, what is the motivation behind it? It being possible in this setting does not mean it is possible in general, so we are unsure what insights such a test gives. This is evidenced by all works we know and cite attempting to do so using learnt approaches that thereby cannot do zero-shot evaluation - if you know of other works that can do this in sensor space we would be happy if you could share them.
>
> As for the technical point - there is a need to match the different channels, not only interpolate some missing ones, as in dataset A channels may have some order and dataset B have a totally different order, even if they had the same sensors in the same locations. This is further exacerbated by the locations in each instance being different, so even if these are the same sensors their information content is different as they measure the magnetic fields at different points. There are model-agnostic ways to try alleviating this, eg. interpolating the values of one dataset’s sensors based on the locations of the other’s, but we are unaware of works that do this and, unless there are techniques we are unaware of, would call this generically nontrivial. Moreover, more conceptually, if such a technique existed it would not subtract from source space’s other listed benefits.
>
> **References:**
>
> [1] Défossez, Alexandre, et al. "Decoding speech perception from non-invasive brain recordings." Nature Machine Intelligence

---

> > ### Comment · Reviewer_PhyF · 2024-11-23
> >
> > Thank you for the clarification. I could use some help understand why signal before the stimulus can have ~65% accuracy on detection task?

---

> > > ### Author Response · Authors · 2024-11-23
> > >
> > > We believe that's due to the signals being highly correlated - the task is heard speech detection on words that are read from stories or randomly shuffled sentences, depending on the dataset. As there are contiguous periods of speech followed by gaps/silence, offsetting the labels by short periods leads to minimal mislabeling relative to no offset.
> > >
> > > We are keen to hear your feedback on our reply for **C3-C5**, specifically whether it is likely there is a methodological mistake we missed.

---

> > > > ### Comment · Reviewer_PhyF · 2024-11-24
> > > >
> > > > There is nothing wrong with using MNE for source imaging. I am having doubt whether using MNE can lead to improved performance as you claimed for different reasons I already mentioned. I have not seen any new proof/data to support your claim. Especially now that you show the table regarding offset vs performance, I am even more confused regarding what kind of features were extracted by the NN for the classification. It is definitely NOT physiological signals related to speech. In YOUR reference, Table A.5 in [1] showed the model performed best around the evoked potential and it even showed 0-100 ms would worsen the performance if included as input. Your table showed the model can predict things before it even happened, and signals before the evoked potential lead to the best performance. How can I trust this counter-intuitive result.
> > > >
> > > > [1] Défossez, Alexandre, et al. "Decoding speech perception from non-invasive brain recordings." Nature Machine Intelligence

---

> > > > > ### Author Response · Authors · 2024-11-26
> > > > >
> > > > > Regarding the label offset ablation - please note our explanation from earlier. Due to much of the data consisting of contiguous blocks of speech/silence, the offset likely leads to minimal mislabelling, especially as it is a binary classification task. Hence, we do not think that this ablation shows any deep problems, and if anything is in line with [1] who also saw that shifting their labels (for a completely different task and one that has epoched data as input) has a negligible impact on their model's performance.
> > > > >
> > > > > Regarding MNE - we thought this was covered by the discussion we had earlier on comments **C3-C5**. Why is this coupled with the empirical results unconvincing? We copy the main relevant comment here for convenience.
> > > > >
> > > > > "C3–C5: Ah, we believe we understand now. Just to make sure - you mean that tomographic/global methods, such as MNE, smear the resulting activity so it is akin to a blurred image instead of one with sharp details, where CNN-like models that roughly rely on hierarchical edge detectors are better suited for the latter instead of the former? If so, while we agree that intuitively local methods (eg. an LCMV beamformer) would be better suited then, as per the ablations detailed in section 2 and Appendix B it performs worse. Although better tuned source reconstruction techniques might work even better, this does not nullify the CNN’s empirical success given the current setup. This is intuitive - even if the input is analogous to a blurred image, a CNN would perform better on it than an MLP.
> > > > >
> > > > > As for the SNR, how did you reach the conclusion that our setup does not improve SNR? Please see comments and questions in the next reply. Moreover and more importantly, how would this invalidate any of the given results?"

---

### Official Review · Reviewer_Nwcv · 2024-11-03

**Soundness:** 2
**Presentation:** 2
**Contribution:** 1
**Rating:** 3
**Confidence:** 4

**Summary:**

This paper concerns the interesting question of sensor-space vs source-space decoding of neural signals (MEG/EEG). The hypothesis is that source-space decoding can provide a number of benefits including design of spatial inductive biases, spatial data augmentations, better interpretability, zero-shot generalisation between datasets, and data harmonisation.
The paper is based on two MEG data set and find evidence in favor of the hypotheses

**Strengths:**

The experimental design is relevant to the question.
 Pipelines based on the source and sensor-space representations are carefully optimized for hyperparameters.
A number of data augmentation strategies are detailed.
There is an attempt at explainable AI.

**Weaknesses:**

1)	The question of decoding source and sensor space for M/EEG is not novel, key works include
Edelman et al. 2015. EEG source imaging enhances the decoding of complex right-hand motor imagery tasks. IEEE Transactions on Biomedical Engineering, 63(1), pp.4-14.  (300+ citations)
Andersen et al. 2017, March. EEG source imaging assists decoding in a face recognition task. In 2017 IEEE International Conference on Acoustics, Speech and Signal Processing (ICASSP) pp. 939-943.
Li et al. 2021. A novel decoding method for motor imagery tasks with 4D data representation and 3D convolutional neural networks. Journal of Neural Engineering, 18(4), p.046029.
Leung et al. 2024. Limited value of EEG source imaging for decoding hand movement and imagery in youth with brain lesions. Brain-Computer Interfaces, 11(3), pp.143-157.

2)	While the authors make valid attempts to place the hypotheses and experiments in a real-world/applied context, the core arguments are misleading. For example it is stated that fMRI is acquired in 3D, it is not. The 3D fMRI signal is reconstructed from a k-space measurement in many ways resembling the source reconstructed signals (M/EEG). Similarly, the argument that source space representation enables data augmentation is weak. E.g the possibility of decorrelating sensors (Laplacian filter) would enable a simple localized “sensor masking process”.

3)	Simple baselines are missing, eg. basic classifiers(logreg, SVMs etc). While it is correct that source space is useful for integrating data sets with different spatial sensos locations, extant work use simple heuristics for sensor matching, see e.g. Kostas, D., Aroca-Ouellette, S. and Rudzicz, F., 2021. BENDR: Using transformers and a contrastive self-supervised learning task to learn from massive amounts of EEG data. Frontiers in Human Neuroscience, 15, p.653659.

4)	Explainable learning from M/EEG is a rich research field and references are missing, e.g. the survey: Zhou, X., Liu, C., Zhai, L., Jia, Z., Guan, C. and Liu, Y., 2023. Interpretable and robust ai in eeg systems: A survey. arXiv preprint arXiv:2304.10755.

**Questions:**

Have you considered simple baselines, e.g. SVMs? It is unclear how the differences in representation dimensions (sensor vs space) interact with large MLPs

	Would it be possible to enlarge the set of experiments, there are numerous open source EEG data sets. Standard brain models could be used for source space reconstruction (many data sets do not include structural data)

	How do you compute the probabilities in Table 5 and Table 8?

---

> ### Author Response · Authors · 2024-11-15
>
> Thanks for the feedback and many pointers. Here are some comments, explanations, and questions on it.
>
> As for the comments:
>
> **C1:** The question of decoding source and sensor space for M/EEG is not novel, key works include Edelman et al. 2015\. EEG source imaging enhances the decoding of complex right-hand motor imagery tasks. IEEE Transactions on Biomedical Engineering, 63(1), pp.4-14. (300+ citations) Andersen et al. 2017, March. EEG source imaging assists decoding in a face recognition task. In 2017 IEEE International Conference on Acoustics, Speech and Signal Processing (ICASSP) pp. 939-943. Li et al. 2021\. A novel decoding method for motor imagery tasks with 4D data representation and 3D convolutional neural networks. Journal of Neural Engineering, 18(4), p.046029. Leung et al. 2024\. Limited value of EEG source imaging for decoding hand movement and imagery in youth with brain lesions. Brain-Computer Interfaces, 11(3), pp.143-157.
>
> **A1:** This study does not purport to be the first to decode from either source or sensor space, did you feel this was not clear from the paper? If so I would be happy if you could point to where you felt such an assertion was made. For sensor space lines 33-45 have several citations to existing work, with more works brought up where relevant later on in the paper. Source space is similar, with lines 75-81 giving a few examples of previous works.
>
> These previous works only further support why source space should be studied \- there have been some successes with it but in large scale neural decoding most still use sensors as inputs. Most if not all of these works look at source space in isolation.
>
> Lines 82-88 outline this work’s aim \- rigorously comparing sensor and source space decoding while showing the benefits of source space, most of which are hard if not impossible to get when using just sensor data. As the paper is currently presented, is this unclear?
>
> ---
> **C2:** While the authors make valid attempts to place the hypotheses and experiments in a real-world/applied context, the core arguments are misleading. For example it is stated that fMRI is acquired in 3D, it is not. The 3D fMRI signal is reconstructed from a k-space measurement in many ways resembling the source reconstructed signals (M/EEG). Similarly, the argument that source space representation enables data augmentation is weak. E.g the possibility of decorrelating sensors (Laplacian filter) would enable a simple localized “sensor masking process”.
>
> **A2:** Thanks for refining the point about fMRI, we will change it to be more precise. Could you please outline cases where you feel the paper is misleading? We tried being as scientifically honest as one can be but perhaps it needs to be made clearer.
>
> As for the fMRI point, while technically true we do not think this significantly affects the motivation or results as it is some of the general background on non-invasive methods. Please do share constructive criticism if you believe we are missing something.
>
> Regarding the possibility of spatial data augmentation in sensor space, we do not assert that they are impossible. When discussing sensor vs source space in the abstract and lines 29-31 we just say that it is difficult/nontrivial. In lines 42-44 we discuss spatial data augmentations in sensor space, noting that extensive works on them only give simple ones due to the sensors’ odd spatial structure. In 307-310 we explicitly say that there are few spatial augmentations in sensor space \- not none.
>
> Note that decorrelation is not needed for masking. In source space many voxels are likely correlated, hence why cube masking outperforms slice dropout, as mentioned in lines 318-323.
>
> Although these augmentations have not been suggested before and are harder to generically implement than those in source space, we agree that they are worth further investigating. We will add such an augmentation to sensor space in section 4\.

---

> > ### Author Response · Authors · 2024-11-15
> >
> > **C3:** Simple baselines are missing, eg. basic classifiers(logreg, SVMs etc). While it is correct that source space is useful for integrating data sets with different spatial sensos locations, extant work use simple heuristics for sensor matching, see e.g. Kostas, D., Aroca-Ouellette, S. and Rudzicz, F., 2021\. BENDR: Using transformers and a contrastive self-supervised learning task to learn from massive amounts of EEG data. Frontiers in Human Neuroscience, 15, p.653659.
> >
> > **A3:** Note that the work you cited \[1\] projects the data into a shared learnt latent space \- we do not posit that combining datasets is inherently impossible in sensor space, just that it is more difficult and does not generically allow other orthogonal methods, eg. zero-shot cross-dataset generalisation. There are specialised models that would potentially be able to do so in sensor space, which is discussed several times eg. in lines 273-278, 385-387, 464-469. Their disadvantages are given conceptually in lines 183-188 and empirically shown in Table 4, where the sensor GAT, a model that could in principle handle different sensor configurations as inputs, underperforms. In \[1\] they take a constant number of channels and pad if there are not enough, throwing away potentially useful information and illustrating the difficulty with combining datasets in sensor space. While this is a specific example, to the best of our knowledge it is indicative, as are the cases we cite in the paper.
> >
> > Do you know of works that combine datasets in sensor space without these difficulties? If so we would be keen to read them.
> >
> > As for the additional baselines, we partially discuss in lines 75-89 why we focus on deep learning models but will make it clearer. The short version is that models without feature learning a) are significantly affected by the input representation’s dimensionality, which would give source space an unfair advantage, b) only tell how good the data in the input representation is, without taking into account possible correlations, and c) are less relevant given deep learning models generally performing better. Do you feel that these baselines are relevant in spite of this? If so we would be happy to understand why.
> >
> > ---
> > **C4:** Explainable learning from M/EEG is a rich research field and references are missing, e.g. the survey: Zhou, X., Liu, C., Zhai, L., Jia, Z., Guan, C. and Liu, Y., 2023\. Interpretable and robust ai in eeg systems: A survey. arXiv preprint arXiv:2304.10755
> >
> > **A4:** Thanks for the reference, we will add it and expand the discussion on the related work. Hopefully this will help put this work in context.
> >
> > ---
> > **C5:** Have you considered simple baselines, e.g. SVMs? It is unclear how the differences in representation dimensions (sensor vs space) interact with large MLPs
> >
> > **A5:** A3 addresses the first part, as for the second \- this is one of the factors we aimed to eliminate by extensively tuning hyperparameters and having the models have the same number of parameters. We believe the positive results of \[2\] in source space may be due to using simple models, hence being biased towards the input with the larger dimensionality. Given sufficiently powerful models that are fully trained only the inherent information content in each modality should matter, which we believe is a more relevant setting given current works. We’ll make this clearer in the main paper.
> >
> > ---
> > **C6:** Would it be possible to enlarge the set of experiments, there are numerous open source EEG data sets. Standard brain models could be used for source space reconstruction (many data sets do not include structural data)
> >
> > **A6:** We note in lines 133-135, 864-868 that using structurals is generally preferred. But more importantly \- what is insufficient with the existing experiments? And is there a special reason why additional ones should be EEG and not MEG? This would add a nontrivial layer of complexity as it requires tuning a very different sensor preprocessing and source reconstruction pipeline. If such methods become established in deep learning it will be easy to take existing source space dataloaders. This work aims to test and show the benefits of source space moreso than building a comprehensive technical foundation for it, although it does that to some extent as well.

---

> > > ### Author Response · Authors · 2024-11-15
> > >
> > > **C7:** How do you compute the probabilities in Table 5 and Table 8?
> > >
> > > **A7:** The probability of improvement was given in \[3\], for two sets of results A\_1 and A\_2 it is defined as:
> > >
> > > $P\_{A\_1\>A\_2}=\\frac{1}{|A\_1||A\_2|}\\sum\_{a\_1\\in A\_1, a\_2\\in A\_2}{1\_{a\_1\>a\_2}}$
> > >
> > > \[3\] is a well accepted toolbox for statistical tests in deep learning when working with relatively few runs.
> > >
> > > In Table 8 it intuitively means that the worst seed for the combined dataset was better than the best seed for the Schoffelen only dataset.
> > >
> > > **References**
> > > \[1\] Kostas, D., Aroca-Ouellette, S. and Rudzicz, F., 2021\. BENDR: Using transformers and a contrastive self-supervised learning task to learn from massive amounts of EEG data. Frontiers in Human Neuroscience, 15, p.653659.
> > > \[2\] Britta U Westner and Jean-R´emi King. The best of two worlds: Decoding and source-reconstructing m/eeg oscillatory activity with a unique model. bioRxiv, pp. 2023–03, 2023\.
> > > \[3\] Agarwal, Rishabh, et al. "Deep reinforcement learning at the edge of the statistical precipice." *Advances in neural information processing systems* 34 (2021): 29304-29320.

---

> > > ### Comment · Reviewer_Nwcv · 2024-11-25
> > > **Thank you for the clarification and response - maintain conclusion**
> > >
> > > Thank you for the clarifications and engaged response my and other reviewers comments.  I maintain my conclusion, however, that the contribution and findings are too limited to be relevant at ICLR

---

### Official Review · Reviewer_AdfV · 2024-11-04

**Soundness:** 2
**Presentation:** 2
**Contribution:** 2
**Rating:** 3
**Confidence:** 3

**Summary:**

By employing established techniques to reconstruct neural activity from MEG sources into voxel representations, the study demonstrates advantages such as spatial inductive biases, spatial data augmentations, improved interpretability, zero-shot generalization across datasets, and enhanced data harmonization.

**Strengths:**

- The study conducts comprehensive experiments demonstrating that converting surface brain signals into source space provides a more effective input representation, facilitating neural decoding.

**Weaknesses:**

-  This study is primarily exploratory, focusing on the differences between various input forms. As a result, the technical contributions may appear limited to readers in the ICLR community. This paper might be better suited for publication in a more specialized journal within this field.
- The organization of this paper is difficult to follow, which might due to the absence of subtitles (like for the Dataset/Method/Implement detail ...). Additionally, a clearer structure would enhance the overall readability and coherence of the paper, allowing for a more straightforward understanding of the study's objectives and findings.

**Questions:**

Please see the weakness.

---

> ### Author Response · Authors · 2024-11-15
>
> Thanks for the review, we appreciate that it is harder to interact with a non-mainstream paper like ours than more classical ICLR works. Regarding the two comments:
>
> **C1:** This study is primarily exploratory, focusing on the differences between various input forms. As a result, the technical contributions may appear limited to readers in the ICLR community. This paper might be better suited for publication in a more specialized journal within this field.
>
> **Q1:** Specifically regarding applied neuroscience work - [3, 4] are two spotlight works from ICLR 2024, centred on better models/methods for brain computer interfaces and neuroimaging.
>
> In general, of course a smaller percent of people in ICLR may interact with this work as would in, for example, a neuroscience conference. However, ICLR and broadly the big 3 (also NeurIPS and ICML) have historically been very interdisciplinary, evidenced by their call for papers ([https://iclr.cc/Conferences/2025/CallForPapers](https://iclr.cc/Conferences/2025/CallForPapers), which includes “applications to neuroscience & cognitive science”) and to some extent the wide array of workshops, ranging from AI and law to climate to a myriad of other topics.
>
> As for the technical contribution \- this is, in the machine learning sense, an applied work, as it shows benefits of an established method in a novel context by comparing it to the standard. Many such works have been presented at similar venues, eg. DiffDock \[1\] is a seminal paper on machine-learning-based protein docking that was presented at ICLR 2023\. Its main contribution can be summarised as “it’s better to treat protein docking as a generative rather than a regression task”.
>
> Do you have any suggestions as to how to make it better suited for a machine learning audience? Although it requires some domain knowledge, the methods and majority of the work here are in machine learning.
>
> ---
> **C2:** The organization of this paper is difficult to follow, which might due to the absence of subtitles (like for the Dataset/Method/Implement detail ...). Additionally, a clearer structure would enhance the overall readability and coherence of the paper, allowing for a more straightforward understanding of the study's objectives and findings.
>
> **Q2:** This is an important point we debated a lot while writing the paper. We felt that a “classic” structure, while straightforward, would also lead to a lot of confusion as it would require many forward/backward references between sections, eg. the source reconstruction pipeline is ablated using deep learning models, where separating it into a “source reconstruction” and “methods” section would have led to them referencing each other constantly. There are seminal works without such a structure, eg. \[2\], that with good writing make it work, as we aspire to have here. Could you please detail what specifically makes the paper difficult to read as it is?
>
> ---
> You gave a so-so score for soundness and contribution while the mentioned weaknesses are only for clarity. While presentation is important the scientific content is arguably even moreso \- what did you find scientifically unsound or lacking contribution-wise?
>
> **References**
> \[1\] Corso, Gabriele, et al. "Diffdock: Diffusion steps, twists, and turns for molecular docking." *arXiv preprint arXiv:2210.01776* (2022).
>
> \[2\] Hinton, Geoffrey. "Distilling the Knowledge in a Neural Network." *arXiv preprint arXiv:1503.02531* (2015).
>
> [3] Jiang, Wei-Bang, Li-Ming Zhao, and Bao-Liang Lu. "Large brain model for learning generic representations with tremendous EEG data in BCI." arXiv preprint arXiv:2405.18765 (2024).
>
> [4] Ju, Ce, et al. "Deep Geodesic Canonical Correlation Analysis for Covariance-Based Neuroimaging Data." The Twelfth International Conference on Learning Representations.

---

> > ### Comment · Reviewer_AdfV · 2024-11-25
> >
> > **C1**: Thank you for for listing these papers. I also have read these papers [1-2] before and I am fully aware that there are many excellent neuroscience-related (either neuroscience inspired or the application of machine learning in neuroscience) in ML conferences like ICLR, ICML, and NeurIPS. However, these papers (like [1] and [2]) either have solid technical contributions or extensive experiments that validate their findings or demonstrate the effectiveness of the proposed methods in specific neuroscience scenarios. For example, in [1], beyond adapting a transformer-based model to EEG signals and pretraining it on a large corpus of EEG data, the authors validated the learned representations across multiple tasks, such as classification (emotion recognition, abnormality detection, etc) and regression (in their appendix). They also conducted thorough comparisons with other state-of-the-art EEG decoding baselines, further demonstrating the robustness and effectiveness of their approach.
> >
> > The topic of this study—exploring general representations from the source space—is indeed interesting. However, as this work leans more towards an application-oriented or exploratory study rather than presenting novelty on the model side, I would expect more comprehensive and clearly presented experiments to support your claims. For instance, this study focuses solely on performance in speech detection, but your title and abstract are very general. How does the proposed approach perform in other tasks? Including such comparisons could significantly strengthen the impact of the work.
> >
> > **C2**: It is absolutely fine not to adhere strictly to the "classic" structure. That said, under each major section, it would be helpful to introduce more subtitles to break up lengthy texts into smaller, well-defined segments, each focusing on a specific topic. Currently, when reviewing the paper, locating specific details—such as technical explanations or individual results—becomes very challenging because they are mixed together, especially across sections 3 to 7. This structure makes it difficult for readers to summarize your contributions or clearly understand your work. An alternative approach could be to follow a more "classic" structure in the appendix. There, you could systematically present your method—starting with a high-level pipeline overview and then progressing to detailed model design and training procedures. This would provide readers with a reliable reference point to clarify any confusion.
> >
> > Finally, for your last question, as mentioned in C1, it would be beneficial to include more brain decoding tasks in your experiments to strenghen your findings. Additionally, incorporating comparisons with more other baselines could further validate your approach.
> >
> >
> > [1] Jiang, Wei-Bang, Li-Ming Zhao, and Bao-Liang Lu. "Large brain model for learning generic representations with tremendous EEG data in BCI." arXiv preprint arXiv:2405.18765 (2024).
> >
> > [2] Ju, Ce, et al. "Deep Geodesic Canonical Correlation Analysis for Covariance-Based Neuroimaging Data." The Twelfth International Conference on Learning Representations.

---

### Official Review · Reviewer_bKqP · 2024-11-04

**Soundness:** 2
**Presentation:** 2
**Contribution:** 2
**Rating:** 3
**Confidence:** 3

**Summary:**

Authors emphasize the importance of utilizing the 3-dimensional location of sensors and design a method to map sensor-space data to source-space. They state that decoding from MEG/EEG source space has been done but these studies are impractical for real-time decoding or do not use deep learning. They preprocess the data with empirical frequency parameters and determine source reconstruction parameters, that includes subject-specific anatomical scans and a common brain template for multi-subject decoding. They utilize Armeni dataset for single subject experiments and Schoffelen dataset for multi-subject experiments. They compare the source and sensor representations with a 3-Layer MLP architecture, and find close accuracy score in single-subject and multi-subject experiments. They adopt a 3D Convolutional Neural Network (CNN) architecture, where they represent the irregular voxel shape in source-space with the minimal cubic volume, and a graph attention network (GAT), both with a dropout mechanism. Multi-subject experiment shows that the CNN model outperforms MLP and GAT in source and sensor space, and the performance is close to MLP in sensor space. They apply spatial data augmentations; mixup, slice and cube masking, that they claim that there are no spatial augmentations in source space for MEG/EEG data. They experiment on masking out all voxels in a brain region and measure the change in performance, where a consistent trend is not observed. In the zero-shot interdataset generalization experiment, they evaluate between each subject of Armeni dataset, the model trained on multiple subjects of the Schoffelen dataset, and vice versa.  Combining datasets improves single subject results.

**Strengths:**

The promising study introduces a large body of experimental effort on a relatively less explored field, MEG data. The code and pipelines in this work can contribute to increasing interest in the field. Source space reconstruction of MEG data is a solid goal, albeit a well-studied one.  Enabled by the common source template, combining multi-subject datasets improves single subject performance.

**Weaknesses:**

1- The claims, that being the first study to apply CNN architecture or spatial augmentations in the MEG field might be too strong, as some other previous studies [1, 2] worked on very similar goals.

2- Better separating MEG field technicalities and machine learning related details can expand the audience of the work. The article might benefit from a more compact writing style to include figures and definition tables that can guide the reader.

3- As the MEG data is source reconstructed, source space model saliency can be extracted to determine if there are recurring spatial patterns, which is missing in the work.

4- Intuition behind Slice dropout is not well grounded in the article, which can also be defined as a special case of cube masking.

[1] Z. Huang and T. Yu, “Cross-Subject MEG Decoding Using 3D Convolutional Neural Networks,” in 2019 WRC Symposium on Advanced Robotics and Automation (WRC SARA), Aug. 2019, pp. 354–359. doi: 10.1109/WRC-SARA.2019.8931958.

[2] A. Giovannetti et al., “Deep-MEG: spatiotemporal CNN features and multiband ensemble classification for predicting the early signs of Alzheimer’s disease with magnetoencephalography,” Neural Comput & Applic, vol. 33, no. 21, pp. 14651–14667, Nov. 2021, doi: 10.1007/s00521-021-06105-4.

**Questions:**

The study introduces new experiments in the MEG based deep representation learning; zero-shot transfer learning, expanding datasets with a common template. However, a comparison with related studies is missing. Would it be possible to carry out an experiment to compare with related studies in the MEG field?

"Balanced accuracy" concept needs a more detailed explanation.

Time resolution is especially important for the relatively newly explored MEG data in deep learning applications, compared to the spatial resolution of fMRI data. A discussion and improvement of the model architecture towards emphasizing time dimension would be a significant improvement in this study.

---

> ### Author Response · Authors · 2024-11-15
>
> Thanks for your feedback, we would like to address some of your comments and questions. Especially regarding the mentioned weaknesses we would be keen to understand whether they stem from a lack of clarity in the paper.
>
> Comments:
> **C1:** The claims, that being the first study to apply CNN architecture or spatial augmentations in the MEG field might be too strong, as some other previous studies \[1, 2\] worked on very similar goals.
>
> **Q1:** Thanks for referring us to these works, we were unaware of them. We do not assert that we are the first to use CNNs, or generally models that leverage structure, for MEG decoding \- see lines 82-88. This is a benefit of using source space and this study aims to methodically compare sensor to source space, while some specific points have been mentioned or studied in isolation, see lines 75-82. As the de-facto standard input for neural decoding is the sensor data, existing studies seem to have not convinced the wider community that their use of source space is justified. Is this broad conceptual point unclear in the paper as it is?
>
> Minor point  \- note that \[1\] does decoding in sensor space, projecting the sensors onto a 2D grid while \[2\] parcellates the data (divides it into regions of interest). Still these references are certainly relevant and will be incorporated.
>
> ---
>
> **C2:** Better separating MEG field technicalities and machine learning related details can expand the audience of the work. The article might benefit from a more compact writing style to include figures and definition tables that can guide the reader.
>
> **Q2:** This is a good point we extensively deliberated on during the writing. Do you have an example of a paper with such a writing style? We tried to make the paper accessible while maintaining a high scientific quality, drawing on inspiration from seminal deep learning papers that are considered well written in spite of their atypical structures, eg. \[3\].
>
> Could you please point at specific terms which you felt could be better introduced, or more specific points to improve on? That would be much appreciated.
>
> ---
> **C3:** As the MEG data is source reconstructed, source space model saliency can be extracted to determine if there are recurring spatial patterns, which is missing in the work.
>
> **Q3:** Indeed, we deliberated on which kinds of explainability methods to try out. We decided that the region masking presented in section 5 would be the most interpretable and interesting kind of result for domain experts, whereas saliency maps can often yield erroneous correlations that amount to not much more than detecting edges, detailed in \[4\]. We tested saliency maps a bit but admittedly not extensively, and did not pursue them further due to seeing some of the issues discussed in \[4\]. What are the kind of recurring spatial patterns you would expect to see? For interpretability to be more than a “shot in the dark” a hypothesis or at least some broad expectation helps.
>
> ---
> **C4:** Intuition behind Slice dropout is not well grounded in the article, which can also be defined as a special case of cube masking.
>
> **Q4:** Thanks, that’s a good point. We thought the analogy to masking parts of an image is sufficient for both augmentations, is this intuition unclear/insufficient? If so we will add a better explanation.
>
> Minor point \- slice dropout cannot be a special case of cube masking as a) several slices would usually be selected, whereas there would only be one cuboid, , and b) slice dropout has the train vs test time weight tweaking regular dropout has but applied to the input, whereas cube masking has any effect only during training.

---

> > ### Author Response · Authors · 2024-11-15
> >
> > Now as for the questions:
> > **RQ1:** The study introduces new experiments in the MEG based deep representation learning; zero-shot transfer learning, expanding datasets with a common template. However, a comparison with related studies is missing. Would it be possible to carry out an experiment to compare with related studies in the MEG field?
> >
> > **AA1:** As mentioned in lines 210-217 and especially 265-267, it is suboptimal yet somewhat standard to only have internal baselines in MEG decoding. Especially here due to the custom task, data splits, and so on it is hard to compare to other works, further exacerbated by essentially every work having their own special setup. This is alleviated by the baselines being tuned to eliminate confounding factors as much as is possible, having the same number of hyperparameters, hyperparameter tuning, both sensor and source preprocessing pipelines being tuned, etc.
> >
> > If you know of any reasonable comparisons it would be much appreciated if you could share them. Part of the issue of taking existing models and using them with a source instead of sensor space input is that they were tuned to work well on sensor space \- they can be further tuned to work on source space but then it’s potentially unclear if improvements stem from the underlying data or good engineering, which we tried to separate here by using simple setups. As MEG decoding has no standard benchmarks it is difficult to show “this model outperforms all others on task X”, as may be common in fields like computer vision.
> >
> > ---
> > **RQ2:** "Balanced accuracy" concept needs a more detailed explanation.
> >
> > **AA2:** We thought this is a well known term in machine learning, apologies. We will add a brief explanation. Concisely \- it’s the average per-class accuracy, so if a model always predicts one class out of three the balanced accuracy is 33%. As we only consider binary regression random chance or single-class prediction here is 50%.
> >
> > ---
> > **RQ3:** Time resolution is especially important for the relatively newly explored MEG data in deep learning applications, compared to the spatial resolution of fMRI data. A discussion and improvement of the model architecture towards emphasizing time dimension would be a significant improvement in this study.
> >
> > **AA3:** What would discussing/experimenting with temporality add? What do you feel is missing? This work aims to rigorously compare sensor to source space decoding, showing that the latter has some arguably significant benefits the former does not. This is orthogonal to the time dimension, which as it is shared between them and unchanged by source reconstruction is for simplicity not studied here. The spatial resolution is not extensively discussed here but the source space’s spatial structure.
> >
> > ---
> > **References**
> >
> > \[1\] Z. Huang and T. Yu, “Cross-Subject MEG Decoding Using 3D Convolutional Neural Networks,” in 2019 WRC Symposium on Advanced Robotics and Automation (WRC SARA), Aug. 2019, pp. 354–359. doi: 10.1109/WRC-SARA.2019.8931958.
> >
> > \[2\] A. Giovannetti et al., “Deep-MEG: spatiotemporal CNN features and multiband ensemble classification for predicting the early signs of Alzheimer’s disease with magnetoencephalography,” Neural Comput & Applic, vol. 33, no. 21, pp. 14651–14667, Nov. 2021, doi: 10.1007/s00521-021-06105-4.
> >
> > \[3\] Hinton, Geoffrey. "Distilling the Knowledge in a Neural Network." *arXiv preprint arXiv:1503.02531* (2015).
> >
> > \[4\] Adebayo, Julius, et al. "Sanity checks for saliency maps." *Advances in neural information processing systems* 31 (2018).

---

> > ### Comment · Reviewer_bKqP · 2024-11-19
> >
> > **C1 Novelty**: Thanks for the clarification. I had the impression of strong claims (i.e. [L1]) in the study and I gave examples (not asking for reference), since there are previous studies that address components of a 3D space classifier on source reconstructed sensor data. Yet the main claim is to be the first source reconstructed 3D space classifier MEG model that can "harmonize" datasets and allows better zero-shot generalization. I acknowledge the concept of the work on MEG data, yet I would prefer to see a MEG modality specific model choice. For instance, on the source reconstructed 4D (1D temporal and 3D spatial) data, one can separately experiment on the number of parameters in each of the dimension groups with a 3+1 dimensional Convolutional model, a common approach in fMRI domain [3, 4], that may show the benefit of the relatively higher temporal resolution of the MEG modality.
> >
> > Minor point: Although what is meant is clear, the use of "inductive biases" term in the paper is not very accurate, as any model has an inductive bias. The goal is to adopt or develop a model with a certain inductive bias, that complies with the modality of the data, i.e. locally correlated voxels, locally correlated time-frames, translation invariance. To match the goals of the paper and its accomplishments, it can be better to name the specific inductive bias/es that you aimed to have in your model.
> >
> > **C2 Writing**: Compared to Hinton's article, due to the differences in the interdisciplinary work with numerous domain related details, it can be harder to follow in the given "classic" style, where the term classic is subjective. Any related journal style (i.e. IEEE PAMI, Wiley HBM) can be adopted for the writing style. I believe the writing can benefit the most from better emphasizing the important details. For instance, source reconstruction (Hämälainen et al., 1993) is the most central concept in the article. A more detailed summary of this step with references to the tuned parameters can improve the flow of the paper. The model input dimensions should be clear before the model section as the details are given with approximation due to variance among subjects. A table can clarify the data specifications, like time, space dimensions and voxel size, which can placed in the appendix and referenced in the second section. Any metric should be explained or cited before its first use, i.e. balanced accuracy.
> >
> > **C3 Interpretability**: Region masking is a good choice for interpretation, however it might be prone to differences in region volumes as well. Gradient based methods can also give insights about the model ([1] advise not to use guided methods, however this advice might also be data dependent [2]). In fMRI domain, a common approach [3] for relatively validating a deep model is through comparing the deep model saliency with a GLM map (linear model saliency) on a basic task with a well-known signal response, which can be repeated in the current study for a small sample size.
> >
> > **C4 Intuition behind Slice dropout**: Mostly, an augmentation method is selected such that it corresponds to a perturbation that results in either potentially a part of the input distribution (i.e. adding noise in computer vision(CV)) or it tests a causal interventation (i.e. adding/removing/translating an object in CV, sensor removal in EEG). For this reason, I wanted to learn whether there is an intuitive reasoning behind slice dropout regarding source reconstruction or it is an empirical observation.
> >
> > [L1] Lines 307-310, "..., there are few spatial augmentations and none that we are aware of in source space".
> >
> > [1] Adebayo, Julius, et al. "Sanity checks for saliency maps." Advances in neural information processing systems 31 (2018).
> >
> > [2] G. Yona and D. Greenfeld, “Revisiting Sanity Checks for Saliency Maps,” Oct. 27, 2021, arXiv: arXiv:2110.14297. doi: 10.48550/arXiv.2110.14297.
> >
> > [3] X. Wang, “Decoding and mapping task states of the human brain via deep learning,” Hum Brain Mapp, vol. 41, no. 6, pp. 1505–1519, Apr. 2020, doi: 10.1002/hbm.24891.
> >
> > [4] Y. Zhao et al., "Four-Dimensional Modeling of fMRI Data via Spatio–Temporal Convolutional Neural Networks (ST-CNNs)," in IEEE Transactions on Cognitive and Developmental Systems, vol. 12, no. 3, pp. 451-460, Sept. 2020, doi: 10.1109/TCDS.2019.2916916.
> >
> > * Articles cited by author names are from the article bibliography.

---

> > > ### Comment · Reviewer_bKqP · 2024-11-19
> > >
> > > **Q1**: It would not have to be a one-to-one comparison with a specific method but an adoption of its main points. After source reconstruction, one can adopt one of the many options in fMRI domain. Due to the limited time, adopting a simple linear classifier (i.e. SVM) as a baseline can be a convenient means to compare against the proposed methods in the study. But even in the fMRI domain, where the minimally preprocessed relatively higher spatial resolution data is readily geometrically aligned on a common template, it is problematic to functionally align data among subjects and datasets. As a suggestion, an ideal solution would be to apply a linear functional alignment [1] method after source reconstruction.
> > >
> > > [1] J. V. Haxby et al., “A Common, High-Dimensional Model of the Representational Space in Human Ventral Temporal Cortex,” Neuron, vol. 72, no. 2, pp. 404–416, Oct. 2011, doi: 10.1016/j.neuron.2011.08.026.
> > >
> > > **Q2**: See the minor point in C1 above.
> > >
> > > **Q3**: As the MEG modality is one of the main points of novelty, it is counter-intuitive to neglect the impact of temporal resolution of MEG data. See the suggestion in C1 above.

---

> ### Author Response · Authors · 2024-11-21
>
> Thanks for the additional comments, they are quite helpful. Regarding each one:
>
> **C1:** Yes, we tried being precise about what we do and don’t do but perhaps that could be better clarified. Regarding the use of CNNs we’ll add the references you suggested (will update the paper after aggregating replies from all reviewers), but believe statement [L1 ] is still accurate. The harmonization and zero-shot generalization are the main benefits of source space but not the only ones.
>
> As for a MEG specific architecture - is there an issue with the existing models? They respect the domain’s structure, eg. the source space 3D CNN having positional embeddings so it is not totally translation invariant, as activities in different regions have different meanings. On adding temporality, MEG’s higher sampling rate is important but orthogonal to what we study here. This benefit exists regardless of the spatial structure, being source or sensor space. It would be interesting to compare decoding using MEG to fMRI but that’s a different line of research, including works like [1].
>
> Regarding inductive biases - that is a good point, thanks, we will update the paper to better reflect it.
>
> ---
>
> **C2:** Thanks for the constructive, precise feedback. A full rewriting that totally changes the format may be impractical for a rebuttal - while potentially requiring rereviewing - but we think your comment on better emphasising some points is well justified in hindsight. We will try doing so.
>
> ---
>
> **C3:** Yes the region sizes are a potential issue, if the results in section 5 showed a larger trend we planned on investigating this further. In [2] it seems like the GLM and DNN give a similar response no? Hence the DNN does not yield additional insight, other than a general sanity check that it does something reasonable. If you think it is an important sanity we are happy to try it, but would not be surprised if it is lackluster due to the same technical reasons mentioned at the end of section 5.
>
> An idea we had following this discussion - one can perform the region masking ablation but mask out larger, constantly sized volumes centred around different areas. That way a) MEG’s lower localisation will be alleviated, and b) all regions will have approximately the same volume. Doing so with ~50 voxels (out of usually 400+, so about 1/8th of the brain) yielded results with clearer trends and significant drops in accuracy, see plot here https://filebin.net/ma3ay9uc4jcwqvox . It's interesting to note how the (para)cingulate gyrus has a significant effect, although it's not traditionally thought to be related to speech processing, with some modern studies drawing connections eg. [3].
>
> ---
>
> **C4:** Thanks for clarifying. Slice dropout was mostly picked as an analogy with augmentations in image feature spaces, where dropout is applied channel wise to promote independence, with some intuition being that a good model should perform well based on signals from various different brain regions instead of over-relying on specific ones. However, we realized that unlike images here the different channels are highly correlated, leading to cube masking which has a similar goal but takes into account correlations in all dimensions. We will clarify this in the main text.
>
> ---
>
> **Q1:** Would the SVM not still be an internal baseline? We are unsure if we fully understood your proposal and which kind of baseline is it - eg. are you proposing a sanity check for the source space models to show that they are well tuned? What’s the baseline’s purpose? Regardless, such baselines without feature learning have limitations when used to compare between sensor and source space, see C3/A3 for reviewer Nwcv.
>
> ---
>
> **Q3:** We do not think or assert that decoding from MEG is novel - see lines 34-44 and [1].
>
> **References**
>
> [1] Benchetrit, Yohann, Hubert Banville, and Jean-Rémi King. "Brain decoding: toward real-time reconstruction of visual perception." arXiv preprint arXiv:2310.19812 (2023).
>
> [2] X. Wang, “Decoding and mapping task states of the human brain via deep learning,” Hum Brain Mapp, vol. 41, no. 6, pp. 1505–1519, Apr. 2020, doi: 10.1002/hbm.24891.
>
> [3] Gennari, Silvia P., et al. "Anterior paracingulate and cingulate cortex mediates the effects of cognitive load on speech sound discrimination." NeuroImage 178 (2018): 735-743.

---

> > ### Comment · Reviewer_bKqP · 2024-11-22
> >
> > **C1**: MEG specific architecture is indeed orthogonal to your work, but it feels counter-intuitive to see a MEG study that neglects the temporal dimension of the data in model properties (although I acknowledge it is not your claim as a point of novelty). Has there been a specific reason behind your approach?
> >
> > **C3**: In [1], experiments have similarities between DNN saliency and GLM maps to showcase the proposed model do not contradict with well-known functional properties in the field, as well as having a superior classification performance. A similar approach can be useful to relatively address spurious classification concerns, which will still not be totally ruled out, due to the difficulty of highly nonlinear DNN interpretability.
> >
> > **Q1**: A linear baseline can be of value as a comparison in both performance and saliency, akin to [1].
> >
> > [1] X. Wang, “Decoding and mapping task states of the human brain via deep learning,” Hum Brain Mapp, vol. 41, no. 6, pp. 1505–1519, Apr. 2020, doi: 10.1002/hbm.24891.

---

### Meta-Review · Area_Chair_XhPs · 2024-12-13

**Metareview:**

This works considers brain decoding in the source space after solving the MEG/EEG inverse problem (eg using MNE). Then deep learning models are trained and evaluated in voxel space.

Reviewers raise concerns about the ML novelty of the work and also the actual performance gains enabled by such an approach compared to more simple baseline ML models.

**Additional Comments On Reviewer Discussion:**

Authors engaged with reviewers who have raised numerous concerns about the novelty of the work and its relevance to the ICLR community. Reviewers have maintained their rating leaning towards a clear rejection.

---

### Decision · Program_Chairs · 2025-01-22

Reject